# SustainDC: Benchmarking for Sustainable Data Center Control

**Avisek Naug**[†], **Antonio Guillen**[†], **Ricardo Luna**[†], **Vineet Gundecha**[†], **Cullen Bash**,
**Sahand Ghorbanpour**, **Sajad Mousavi**, **Ashwin Ramesh Babu**, **Dejan Markovikj**,
**Lekhapriya D Kashyap**, **Desik Rengarajan**, **Soumyendu Sarkar**[†*]

Hewlett Packard Enterprise (Hewlett Packard Labs)

{avisek.naug, antonio.guillen, rluna, vineet.gundecha, cullen.bash, sahand.ghorbanpour,
sajad.mousavi, ashwin.ramesh-babu, dejan.markovikj, lekhapriya.dheeraj-kashyap, desik.rengarajan,
soumyendu.sarkar}@hpe.com

## Abstract

Machine learning has driven an exponential increase in computational demand, leading to massive data centers that consume significant energy and contribute to climate change. This makes sustainable data center control a priority. In this paper, we introduce SustainDC, a set of Python environments for benchmarking multi-agent reinforcement learning (MARL) algorithms for data centers (DC). SustainDC supports custom DC configurations and tasks such as workload scheduling, cooling optimization, and auxiliary battery management, with multiple agents managing these operations while accounting for the effects of each other. We evaluate various MARL algorithms on SustainDC, showing their performance across diverse DC designs, locations, weather conditions, grid carbon intensity, and workload requirements. Our results highlight significant opportunities to improve data center operations using MARL algorithms. Given the increasing use of DC due to AI, SustainDC provides a crucial platform for developing and benchmarking advanced algorithms essential for achieving sustainable computing and addressing other heterogeneous real-world challenges.

## 1 Introduction

One of the growing areas of energy and carbon footprint ($CFP$) can be traced to cloud data centers (DCs). The increased use of cloud resources for batch workloads related to AI model training, multimodal data storage and processing, or interactive workloads like streaming services, hosting websites have prompted enterprise clients to construct numerous data centers. Governments and regulatory bodies are increasingly focusing on environmental sustainability and imposing stricter regulations to reduce carbon emissions. This has prompted industry-wide initiatives to adopt more intelligent DC control approaches. This paper presents SustainDC, a sustainable DC Multi-Agent Reinforcement Learning (MARL) set of environments. SustainDC helps promote and prioritize sustainability, and it serves as a platform that facilitates collaboration among AI researchers, enabling them to contribute to a more environmentally responsible DC.

The main contributions of this paper are the following:

- A highly customizable suite of environments focused on Data Center (DC) operations, designed to benchmark energy consumption and carbon footprint across various DC config-

---

*Corresponding author. †These authors contributed equally.

urations. The framework supports the subclassing of models for different DC components ranging from workloads and individual server specifications to cooling systems, enabling users to test fine-grained design choices.

- The environments are implemented using the Gymnasium *Env* class, facilitating the benchmarking of various control strategies to optimize energy use, reduce carbon footprint, and evaluate related performance metrics.

- Supports both homogeneous and heterogeneous multi-agent reinforcement learning (MARL) controllers and traditional non-ML controllers. Extensive studies within these environments demonstrate the advantages and limitations of various multi-agent approaches.

- SustainDC enables reward shaping, allowing users to conduct ablation studies on specific DC components to optimize performance in targeted areas.

- SustainDC serves as a comprehensive benchmark environment for heterogeneous, multi-agent, and multi-objective reinforcement learning algorithms, featuring diverse agent interactions, customizable reward structures, high-dimensional observations, and reproducibility.

Code, licenses, and setup instructions for SustainDC are available at GitHub[2]. The documentation can be accessed at [3].

## 2 Related Work

Recent advancements in Reinforcement Learning (RL) have led to an increased focus on optimizing energy consumption in areas such as building and DC management. This has resulted in the development of several environments for RL applications. *CityLearn* (1) is an open-source platform that supports single and MARL strategies for energy coordination and demand response in urban environments. *Energym* (2), *RL-Testbed* (3) and *Sinergym* (4) were developed as RL wrappers that facilitate communication between Python and EnergyPlus, enabling RL evaluation on the collection of buildings modeled in EnergyPlus. *SustainGym* (5) is one of the latest suite of general purpose RL tasks for evaluation of sustainability, simulating electric vehicle charging scheduling and battery storage bid, which lends itself to benchmarking different control strategies for optimizing energy, carbon footprint, and related metrics in electricity markets.

Most of the above-mentioned works use *EnergyPlus* (6) or, *Modelica* (7), (8) which were primarily designed for modeling thermo-fluid interactions with traditional analytic control with little focus on Deep Learning applications. The APIs provided in these works only allow sampling actions in a model free manner, lacking a straightforward approach to customization or re-parameterization of system behavior. This is because most of the works have a set of pre-compiled binaries (e.g. FMUs in Modelica) or fine-tuned spline functions (in EnergyPlus) to simulate nominal behavior. Furthermore, there is a significant bottleneck in using these precompiled environments from Energyplus or Modelica for Python based RL applications due to latency associated with cross-platform interactions, versioning issues in traditional compilers for EnergyPlus and Modelica, unavailability of open source compilers and libraries for executing certain applications.

SustainDC allows users to simulate the electrical and thermo-fluid behavior of large DCs directly in Python. Unlike other environments that rely on precompiled binaries or external tools, SustainDC is easily end-user customizable and fast It enables the design, configuration, and control benchmarking of DCs with a focus on sustainability. This provides the ML community with a new benchmark environment specifically for Heterogeneous MARL in the context of DC operations, allowing for extensive goal-oriented customization of the MDP transition function, state space, actions space, and rewards.

## 3 Data Center Operational Model

Figure 1 illustrates the typical components of a DC operation as modeled in SustainDC. *Workloads* are uploaded to the DC from a proxy client. For non-interactive batch workloads, some of these jobs can be scheduled flexibly, allowing delays to different periods during the day for optimization. This

---

[2]GitHub repository: https://github.com/HewlettPackard/dc-rl.

[3]Documentation: https://hewlettpackard.github.io/dc-rl.

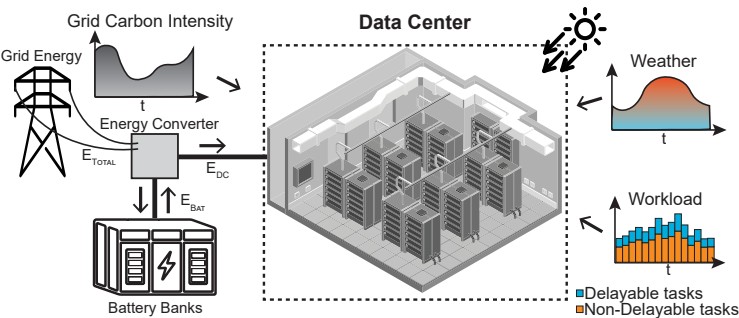

Figure 1: Operational Model of a SustainDC Data Center

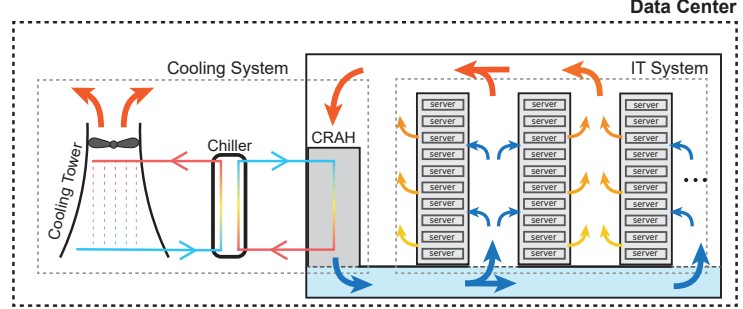

Figure 2: Model of the data center. The configuration allows customization of the number of cabinets per row, the number of rows, and the number of servers per cabinet. The cooling system, comprising the CRAH, chiller, and cooling tower, manages the heat generated by the IT system.

creates a scheduling challenge of postponing workloads to times when *Grid Carbon Intensity* (*CI*), energy consumption, or energy pricing is lower.

As the servers (IT systems) in the DC process these workloads, they generate heat that must be removed. A complex HVAC system with multiple components is used to cool the IT system. As shown in Figure 2, warm air rises from the servers via convection. Driven by the HVAC fan's forced draft, this warm air enters the *Computer Room Air Handler* (CRAH) (depicted by red arrows), where it is cooled to an optimal setpoint by a heat exchange process using a "primary" chilled water loop. The chilled air is then returned to the IT room through a plenum located beneath the DC (shown by blue arrows). The warmed water from this loop returns to the *Chiller*, where another heat exchange process transfers heat to a "secondary" chilled water loop, which carries the heat to a *Cooling Tower*. The cooling tower fan, operating at variable speeds, rejects this heat to the external environment, with fan speed and energy consumption determined by factors such as the secondary loop's inlet temperature at the cooling tower, the desired outlet temperature setpoint, and external air temperature and humidity. Depending on the external *Weather* and processed *Workload*, the IT and cooling systems consume *Grid Energy*. Selecting the optimal cooling setpoint for the CRAH can reduce the DC's carbon footprint and also impacts the servers' energy efficiency (9).

Larger DCs may include onsite *Battery Banks* that charge from the grid during low *CI* periods and may optionally provide auxiliary energy during high *CI* periods. This introduces a decision-making sustainability challenge to determine the optimal charging and discharging intervals for the batteries.

These three control problems are interrelated, motivating the development of testbeds and environments for evaluating multi-agent control approaches that collectively aim to minimize carbon footprint, energy and water usage, energy cost, and other sustainability metrics of interest.

# 4   SustainDC environment overview

A high-level overview of SustainDC is provided in Figure 3, outlining the three main environments developed in *Python* along with their individual components, customization options, and associated control challenges.

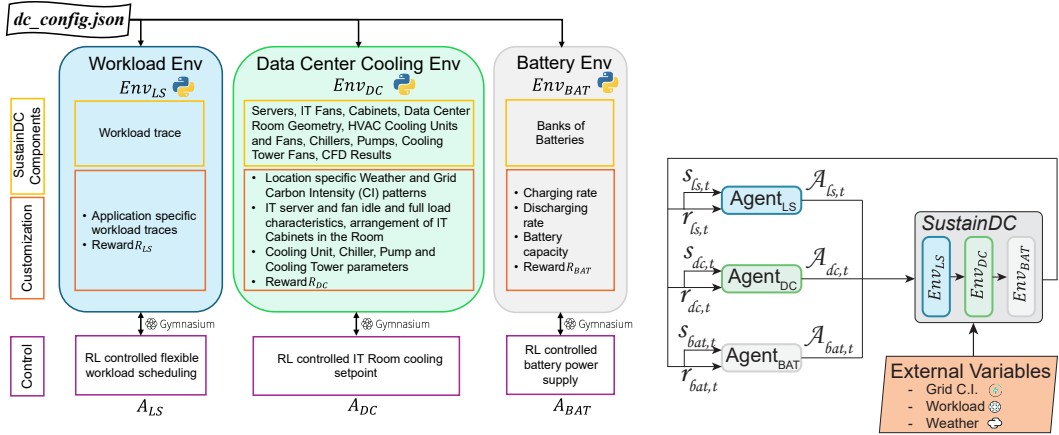

(a) High-level overview of SustainDC, showing the three main environments (*Workload Env*, *Data Center Cooling Env*, and *Battery Env*) along with their customizable components and control actions.

(b) RL loop in SustainDC, depicting how states and actions are formed from individual agents.

Figure 3: SustainDC overview and RL loop

The *Workload Environment* models and controls the execution and scheduling of delay-tolerant workloads within the DC.

In the *Data Center Environment*, servers housed in IT room cabinets process these workloads. This environment simulates both electrical and thermo-fluid dynamics, modeling heat generated by the workload processing and its transfer to the external environment through HVAC cooling components.

The *Battery Environment* simulates grid charging during off-peak hours and supplies auxiliary energy to the DC during periods of high grid carbon intensity, offering a solution to manage energy demand sustainably.

Detailed physics-based implementations for each environment are available in the supplementary document. Customization parameters for all aspects of the DC environment design in SustainDC can be fully specified through *dc_config.json*, a universal configuration file.

Figure 3a further illustrates SustainDC, showing the *Workload Environment*, *Data Center Environment*, and *Battery Environment* along with their customizable parameters. Figure 3b depicts the RL loop in SustainDC, illustrating how agents' actions and states optimize DC operations, considering external variables like grid CI, workload, and weather.

## 4.1 Workload Environment

The *Workload Environment* ($Env_{LS}$) manages the execution and scheduling of delay tolerant workloads within the DC by streaming workload traces (measured in FLOPs) over a specified time period. SustainDC includes a set of open-source workload traces from *Alibaba* (10) and *Google* (11) data centers. Users can customize this component by adding new workload traces to the *data/Workload* folder or by specifying a path to existing traces in the *dc_config.json* file.

Some workloads are flexible, meaning they can be rescheduled within an allowable time horizon. Tasks such as updates or backups do not need immediate execution and can be delayed based on urgency or Service-Level Agreements (SLA). This flexibility allows workloads to be shifted to periods of lower grid carbon intensity (CI), thereby reducing the DC's overall carbon footprint ($CFP$).

Users can also customize the CI data. By default, we provide a one-year CI dataset for the following states: Arizona, California, Georgia, Illinois, New York, Texas, Virginia, and Washington, locations selected due to their high data center density. The carbon intensity data files, sourced from eia.gov (`https://api.eia.gov/bulk/EBA.zip`), are located in the *data/CarbonIntensity* folder.

Let $B_t$ be the instantaneous DC workload trace at time $t$, with $X\%$ of the load reschedulable up to $N$ simulation steps into the future. The objective of an RL agent ($Agent_{LS}$) is to observe the current time of day ($SC_t$), the current and forecast grid CI data ($CI_{t...t+L}$), and the remaining amount of

reschedulable workload ($D_t$). Based on these observations, the agent chooses an action $A_{ls,t}$ (as shown in Table 1) to reschedule the flexible portion of $B_t$, to minimize the net $CFP$ over $N$ steps.

## 4.2 Data Center Environment

The *Data Center* environment ($Env_{DC}$) provides a comprehensive set of configurable models and specifications. For IT-level design, SustainDC enables users to define IT Room dimensions, server cabinet arrangements (including the number of *rows* and *cabinets* per row), and both *approach* and *return* temperatures. Additionally, users can specify server and fan power characteristics, such as *idle power*, *rated full load power*, and *rated full load frequency*.

On the cooling side, SustainDC allows customization of the *chiller reference power*, *cooling fan reference power*, and the supply air *setpoint* temperature for IT Room cooling. It also includes specifications for the pump and cooling tower, such as *rated full load power* and *rated full load frequency*. All these parameters can be configured in the *dc_config.json* file.

One of SustainDC's key features is its ability to automatically adjust HVAC cooling capacities based on workload demands and IT room configurations, a process known as "sizing." This ensures that the data center remains adequately cooled without unnecessary energy expenditure. In contrast, previous environments often neglected this capability, resulting in inaccurate outcomes. For example, changing IT room configurations in other environments typically impacted only IT energy consumption without considering the overall cooling requirements, leading to inconsistent RL-based control results, as seen in *RL-Testbed* in (3). SustainDC addresses this by integrating custom supply and approach temperatures derived from Computational Fluid Dynamics (CFD) simulations, simplifying the complex calculations of temperature changes between the IT Room HVAC and the IT Cabinets (9).

In addition, SustainDC includes weather data (in *data/Weather*) in the .epw format for the same locations as the CI data. This data, sourced from `https://energyplus.net/weather`, represents typical weather conditions for these regions. Users can also specify their own weather files if needed.

Given $\hat{B}_t$ as the adjusted workload from the *Workload Environment*, the goal of the RL agent ($Agent_{DC}$) is to select an optimal cooling setpoint $A_{dc,t}$ (Table 1) to minimize the net carbon footprint $CFP$ from combined cooling ($E_{hvac}$) and IT ($E_{it}$) energy consumption over an $N$-step horizon. In SustainDC, the agent's default state space includes the time of day and year ($SC_t$), ambient weather ($t_{db}$), IT Room temperature ($t_{room}$), previous step cooling ($E_{hvac}$) and IT ($E_{it}$) energy usage, and forecasted grid CI data ($CI_{t...t+L}$).

## 4.3 Battery Environment

The *Battery Environment* ($Env_{BAT}$) is based on battery charging and discharging models, such as $f_{charging}(BatSoc, \delta\tau)$ from (12). Parameters for these components, including battery capacity, can be configured in the *dc_config.json* file.

The objective of the RL agent ($Agent_{BAT}$) is to optimally manage the battery's state of charge ($BatSoc_t$). Using inputs such as the net energy consumption ($E_{hvac} + E_{it}$) from the *Data Center* environment, the time of day ($SC_t$), the current battery state of charge ($BatSoc_t$), and forecasted grid CI data ($CI_{t...t+L}$), the agent determines an action $A_{bat,t}$ (as outlined in Table 1). Actions include charging the battery from the grid, taking no action, or discharging to provide auxiliary energy to the data center, all aimed at minimizing the overall carbon footprint, energy consumption, etc.

## 4.4 Heterogeneous Multi Agent Control Problem

While SustainDC enables users to tackle the individual control problems for each of the three environments, the primary goal of this paper is to establish a multi-agent control benchmark that facilitates joint optimization of the $CFP$ by considering the coordinated actions of all three agents ($Agent_{LS}$, $Agent_{DC}$, and $Agent_{BAT}$). The sequence of operations for the joint multi-agent and multi-environment functions can be represented as follows:

$$Agent_{LS} : (SC_t \times CI_t \times D_t \times B_t) \rightarrow A_{ls,t} \quad (1)$$

$$Agent_{DC} : (SC_t \times t_{db} \times t_{room} \times E_{hvac} \times E_{it} \times CI_t) \rightarrow A_{dc,t} \quad (2)$$

$$Agent_{BAT} : (SC_t \times Bat\_SoC \times CI_t) \rightarrow A_{bat,t} \quad (3)$$

$$Env_{LS} : (B_t \times A_{ls,t}) \rightarrow \hat{B}_t \quad (4)$$

$$Env_{DC} : (\hat{B}_t \times t_{db} \times t_{room} \times A_{dc,t}) \rightarrow (E_{hvac}, E_{it}) \quad (5)$$

$$Env_{BAT} : (Bat\_SoC \times A_{bat,t}) \rightarrow (Bat\_SoC, E_{bat}) \quad (6)$$

$$CFP_t = (E_{hvac} + E_{it} + E_{bat}) \times CI_t \quad (7)$$

where $E_{bat}$ represents the net discharge from the battery based on the change in battery state of charge ($Bat\_SoC$), which can be positive or negative depending on the action $A_{bat,t}$. If the battery provides auxiliary energy, $E_{bat}$ is negative; if it charges from the grid, $E_{bat}$ is positive.

The objective of the multi-agent problem is to determine $\theta_{LS}$, $\theta_{DC}$, and $\theta_{BAT}$, which parameterize the policies for $Agent_{LS}$, $Agent_{DC}$, and $Agent_{BAT}$, respectively, such that the total $CFP$ is minimized over a specified horizon $N$. For this study, we set $N = 31 \times 24 \times 4$, representing a 31-day horizon with a step duration of 15 minutes.

$$\left( \theta_{LS}, \theta_{DC}, \theta_{BAT} \right) = argmin \left( \sum_{t=0}^{t=N} CFP_t \right) \quad (8)$$

## 4.5 Rewards

While $CFP$ reduction is the default objective in SustainDC, the reward formulation is highly customizable, allowing users to incorporate alternative objectives such as total energy consumption, operating costs across all DC components, and water usage.

We primarily consider the following default rewards for the three environments ($Env_{LS}$, $Env_{DC}$, $Env_{BAT}$):

$$(r_{LS}, r_{DC}, r_{BAT}) = \left( -(CFP_t + LS_{Penalty}), -(E_{hvac,t} + E_{it,t}), -(CFP_t) \right)$$

Here, $LS_{Penalty}$ is a penalty applied to the Load Shifting Agent ($Agent_{LS}$) in the Workload Environment ($Env_{LS}$) if it fails to reschedule flexible workloads within the designated time horizon $N$. Specifically, if $D_t$ is positive at the end of a horizon $N$, $LS_{Penalty}$ is assigned. Details on calculating

| Agent | Control Knob | Actions | | Optimization Strategy | Figure |
|-------|-------------|---------|---|----------------------|--------|
| **Agent$_{\text{LS}}$** | Delay-tolerant workload scheduling | 0 
 1 
 2 | Store Delayable Tasks 
 Compute All Immediate Tasks 
 Maximize Throughput | Shift tasks to periods of lower CI/lower external temperature/other variables to reduce the $CFP$. |  |
| **Agent$_{\text{DC}}$** | Cooling Setpoint | 0 
 1 
 2 | Decrease Setpoint 
 Maintain Setpoint 
 Increase Setpoint | Optimize cooling by adjusting cooling setpoints based on workload, external temperature, and CI. |  |
| **Agent$_{\text{BAT}}$** | Battery energy supply/store | 0 
 1 
 2 | Charge Battery 
 Hold Energy 
 Discharge Battery | Store energy when CI/temperature/workload/other is low and use stored energy when is high to reduce $CFP$. |  |

Table 1: Overview of control choices in SustainDC: the tunable knobs, the respective action choices, optimization strategies, and visual representations.

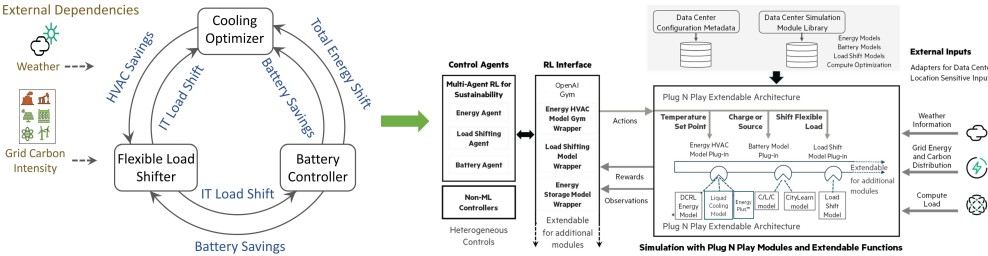

Figure 4: Extendable and plug-and-play design of SustainDC for data center control to address the multi-agent holistic optimization of data centers for resolving multiple dependencies in real-time.

$LS_{Penalty}$ are provided in the supplemental document. Users can opt for custom reward formulations by subclassing the base reward class in *utils/reward_creator.py*.

Based on these individual rewards, we can formulate an independent or collaborative reward structure, where each agent receives partial feedback in the form of rewards from the other agent-environment pairs. The collaborative feedback reward formulation for each agent is formulated as:

$$R_{LS} = \alpha * r_{LS} + (1 - \alpha)/2 * r_{DC} + (1 - \alpha)/2 * r_{BAT}$$
$$R_{DC} = (1 - \alpha)/2 * r_{LS} + \alpha * r_{DC} + (1 - \alpha)/2 * r_{BAT}$$
$$R_{BAT} = (1 - \alpha)/2 * r_{LS} + (1 - \alpha)/2 * r_{DC} + \alpha * r_{BAT}$$

Here, $\alpha$ is the weighting parameter. This reward-sharing mechanism enables agents to incorporate feedback from their actions across environments, making it suitable for independent critic multi-agent RL algorithms, such as IPPO (13). For instance, the adjusted CPU load $\hat{B}_t$ influences data center energy demand ($E_{cool} + E_{it}$), which subsequently affects the battery optimizer's charge-discharge decisions and ultimately impacts the net $CO_2$ footprint. Consequently, we explore a collaborative reward structure and conduct ablation experiments with varying $\alpha$ values to assess the effectiveness of reward sharing.

## 4.6 Extendable plug-n-play Data Center Simulation Platform

Figure 4 illustrates the extendable and plug-and-play design of SustainDC framework for data center control to address the multi-agent optimization of data centers for resolving multiple internal and external dependencies of agents in real-time. The three different controllers for **Cooling Optimizer, Flexible Load Shifter** and **Battery Controller** can be substituted with **RL** or **non-RL controllers**. Similarly, the underlying models performing the simulation can be substituted easily using the **Modules and Extendable Functions** block. In the future, we plan to include the models for next generation of fanless direct liquid cooling for AI servers (14) for Energy HVAC Model Plug-in.

## 5 Evaluation Metrics and Experimental Settings

We consider five metrics to evaluate various RL approaches on SustainDC. The $CO_2$ *footprint* ($CFP$) represents the cumulative carbon emissions associated with DC operations over the evaluation period. *HVAC Energy* refers to the energy consumed by cooling components, including the chiller, pumps, and cooling tower. *IT Energy* refers to the energy consumed by the servers within the DC. *Water Usage*, the volume of chilled water recirculated through the cooling system, is a critical metric in DCs where chilled water supply from a central plant is constrained, and efficient use of this resource helps minimize the DC's water footprint. Finally, *Task Queue* tracks the accumulation of compute FLOPs from workloads that are deferred for rescheduling under lower CI periods. Higher Task Queue values indicate poorer SLA performance within the DC.

Experiments were conducted on an Intel® Xeon® Platinum 8470 server with 104 CPUs, utilizing 4 threads per training agent. All hyperparameter configurations for benchmark experiments are detailed in the supplemental document. The codebase and documentation are linked to the paper.

# 6 Benchmarking Algorithms on SustainDC

The purpose of SustainDC is to explore the benefits of jointly optimizing the *Workload*, *Data Center*, and *Battery Environments* to reduce the operating $CFP$ of a DC. To investigate this, we can perform ablation studies in which we evaluate net operating $CFP$ by running trained RL agents in only one or two of the SustainDC environments while employing baseline methods ($B_*$) in the other environments. For the *Workload Environment* ($Env_{LS}$), the baseline ($B_{LS}$) assumes no workload shifting over the horizon, which aligns with current standard practices in most data centers. For the *Data Center Environment* ($Env_{DC}$), we use the industry-standard ASHRAE Guideline 36 as the baseline ($B_{DC}$) (15). In the *Battery Environment* ($Env_{BAT}$), we adapt the method from (12) for real-time operation, reducing the original optimization horizon from 24 hours to 3 hours as our baseline ($B_{BAT}$). Future work will include further baseline comparisons using Model Predictive Control (MPC) and other non-ML control algorithms.

Next, we perform ablations on the collaborative reward parameter $\alpha$, followed by benchmarking various multi-agent RL approaches. This includes multi-agent PPO (16) with an independent critic for each actor (IPPO) (13) and a centralized critic with access to states and actions from other MDPs (MAPPO) (17). Given the heterogeneous nature of action and observation spaces in SustainDC, we also benchmark several heterogeneous multi-agent RL (HARL) methods (18), including HAPPO (Heterogeneous Agent PPO), HAA2C (Heterogeneous Agent Advantage Actor Critic), HAD3QN (Heterogeneous Agent Dueling Double Deep Q Network), and HASAC (Heterogeneous Agent Soft Actor Critic). MARL agents were trained on one location and evaluated across different locations.

In Figure 5, we compare the relative performance of different RL algorithms using a radar chart based on the evaluation metrics in Section 5. Since reporting absolute values may lack context, we instead plot relative performance differences, offering insights into the *pros* and *cons* of each approach. (Absolute values for these benchmark experiments are provided in the supplementary document in tabular format.) Metrics are normalized by their mean and standard deviation, with lower values positioned at the radar chart periphery and higher values toward the center. Hence, the larger the area for an approach on the radar chart, the better its performance across the evaluated metrics.

## 6.1 Single vs multi-agent Benchmarks

Figure 5a compares the relative performance of a single RL agent versus multi-agent RL benchmarks, highlighting the advantages of a MARL approach for sustainable DC operations. Among single RL agent approaches, the workload manager RL agent (Experiment 1) and the battery agent (Experiment 3) perform similarly in reducing water usage. The standalone DC (cooling) RL agent (Experiment 2) demonstrates strong performance in both energy and $CFP$ reduction. Note that for Experiments 1 and 3, the Lowest Task Queue metric should be disregarded, as the baseline workload manager does not shift workloads and thus inherently has the lowest task queue.

When we evaluate pairs of RL agents working simultaneously, the absence of a cooling optimization agent (e.g., Experiment 5) results in performance similar to single RL agent implementations (Experiments 1 and 3), where only $A_{LS}$ or $A_{BAT}$ are used with baseline agents. This indicates that the RL-based cooling optimizer significantly improves overall performance compared to the rule-based ASHRAE Guideline 36 controller (as seen in Experiments 2 and 4). Finally, when all three RL agents operate simultaneously without a shared critic (Experiment 7 using IPPO), they achieve better outcomes in energy consumption, water usage, and task queue management, with a $CFP$ relatively similar to other experiments. The combined performance across all three agents highlights the benefits of a MARL approach for DC optimization.

## 6.2 Reward Ablation on $\alpha$

Figure 5b, shows the relative differences in performance when considering collaborative reward components. We considered 2 values of $\alpha$ at the extremes to indicate no collaboration ($\alpha = 1.0$) and relying only on the rewards of other agents ($\alpha = 0.1$). An intermediate value of $\alpha = 0.8$ was chosen based on similar work on reward-based collaborative approach in (19; 20). The improvement in setting $\alpha = 0.8$ shows that considering rewards from other agents can improve performance w.r.t. no collaboration ($\alpha = 1.0$) especially in a partially observable MDP.

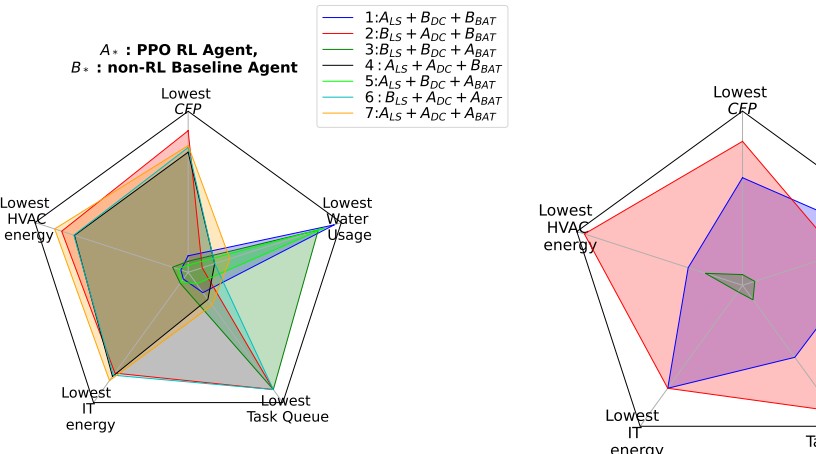

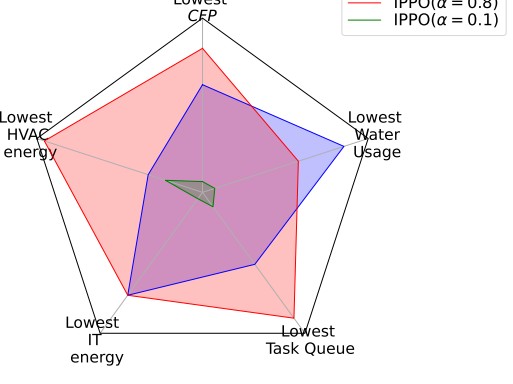

(a) Single RL agent, two RL agents and three RL agents For single agents, PPO was used (Average result over 5 runs)

(b) IPPO with different values of collaborative reward coefficient $\alpha$ (Average result over 12 runs)

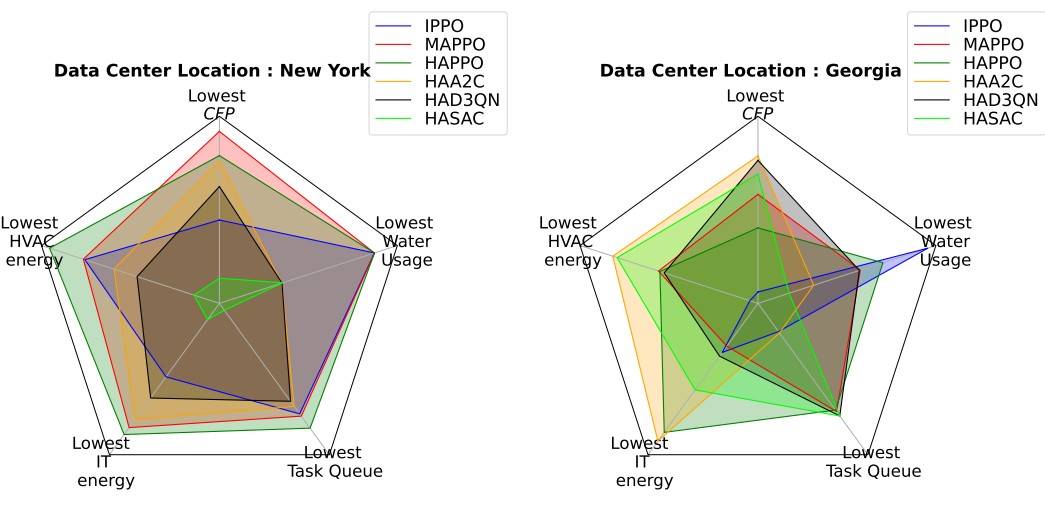

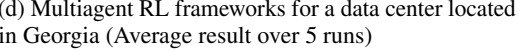

(c) Multiagent RL frameworks for a data center located in New York (Average result over 5 runs)

(d) Multiagent RL frameworks for a data center located in Georgia (Average result over 5 runs)

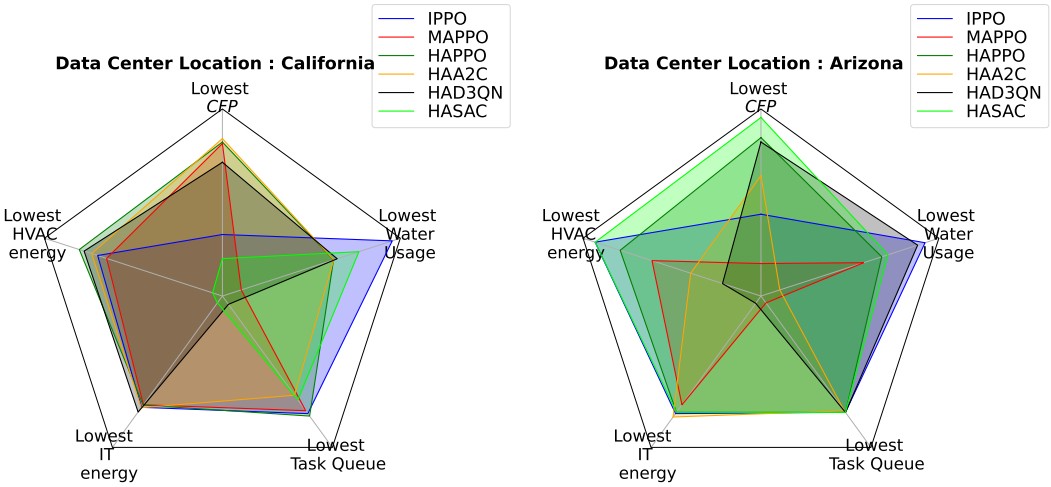

(e) Multiagent RL frameworks for a data center located in California (Average result over 5 runs)

(f) Multiagent RL frameworks for a data center located in Arizona (Average result over 5 runs)

Figure 5: Benchmarking RL Algorithms on the Sustain DC environment

### 6.3 Multiagent Benchmarks

We evaluated and compared the relative performances of various MARL approaches, including PPO with independent actor-critics (IPPO, $\alpha = 0.8$), centralized critic PPO (MAPPO), heterogeneous multi-agent PPO (HAPPO), HAA2C, HAD3QN, and HASAC. Figures 5c, 5d, 5e, and 5f illustrate the relative performance of these methods for DCs located in New York, Georgia, California, and Arizona. Our results reveal a consistent trend where PPO-based shared actor-critic methods (MAPPO, HAPPO) outperform the independent agent counterpart, IPPO. On further analysis, we observed that while IPPO effectively reduces HVAC and IT energy, the battery agent struggles to optimally schedule charging and discharging from the grid to meet data center demand. Among MAPPO, HAPPO, and HAA2C, HAPPO consistently performs best (except in Georgia).

For the off-policy methods (HAD3QN and HASAC), performance varies significantly across regions, with HASAC achieving the highest performance in Arizona. The reasons for these regional performance variations are not fully understood and may be partially due to differences in weather and carbon intensity. We plan to further investigate these variations in future work.

## 7 Limitations

The absence of an oracle that already knows the best results possible for the different environments makes it difficult to quantify the threshold for performance compared to simpler environments. For computational speed in RL, we used reduced order models for certain components like pumps and cooling towers. We could not exhaustably tune the hyperparameters for all the networks.

## 8 Next Steps

We are planning to deploy the trained agents to real data centers and are working towards domain adaptation for deployment with safeguards. We will augment the codebase with these updates. In order to have a smooth integration with current systems where HVAC runs in isolation, we plan a phased deployment with recommendation to the data center operative followed by direct integration of the control agents with the HVAC system with safeguards. For real-world deployment, a trained model should be run on a production server using appropriate checkpoints within a containerized platform with necessary dependencies. Security measures must restrict the software to only read essential data, generate decision variables, and write them with limited access to secure memory for periodic reading by the data center's HVAC management system. To ensure robustness against communication loss, a backup mechanism for generating decision variables is essential.

## 9 Conclusion

This paper introduced SustainDC, a fully Python-based benchmarking environment for multi-agent reinforcement learning (MARL) in sustainable, cost-effective, and energy-efficient data center operations. SustainDC provides comprehensive customization options for modeling multiple aspects of data centers, including a flexible RL reward design, an area we invite other researchers to explore further. We benchmarked an extensive collection of single-agent and multi-agent RL algorithms in SustainDC across multiple geographical locations, comparing their performance to guide researchers in sustainable data center management with reinforcement learning.

Additionally, we are collaborating with consortiums like ExaDigiT, which focuses on high-performance computing (HPC) and supercomputing, as well as with industry partners, to implement some of these approaches in real-world scenarios. SustainDC's complexity and constraints, rooted in realistic systems, make it a suitable platform for benchmarking hierarchical RL algorithms. We plan to implement continual reinforcement learning to accommodate dynamic data center environments and prevent out-of-distribution errors during equipment upgrades and accessory changes.

Moreover, SustainDC features an extendable, plug-and-play architecture of data center modeling compatible with digital twin frameworks, supporting research into other aspects of data center optimization for joint and multi-objective goals.

## Acknowledgement

We would like to thank Paolo Faraboschi for sharing his expertise in machine learning and practical implementation approaches, and Torsten Wilde for his feedback on energy optimization and sustainability.

Additionally, we extend our gratitude to Wes Brewer, Feiyi Wang, Vineet Kumar, Scott Greenwood, Matthias Maiterth, and Terry Jones of Oak Ridge National Laboratory for their feedback and leadership within the ExaDigiT consortium, which helped refine our solution.

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

# APPENDIX

## Contents

Code, licenses, and setup instructions for SustainDC are available at GitHub[4]. The documentation can be accessed at [5].

# A   Models

## A.1   Workload Environment ($Env_{LS}$)

The Workload Environment ($Env_{LS}$) simulates the management and scheduling of data center (DC) workloads, allowing for dynamic adjustment of utilization to optimize energy consumption and carbon footprint. The environment is designed to evaluate the performance of reinforcement learning (RL) algorithms in rescheduling delay-capable workloads within the DC.

Let $B_t$ be the instantaneous DC workload trace at time $t$, with $X\%$ of the load being rescheduled up to $N$ simulation steps into the future. The goal of an RL agent ($Agent_{LS}$) is to observe the current time of day ($SC_t$), the current and forecast grid CI data ($CI_{t...t+L}$), and the amount of rescheduled workload left ($D_t$). Based on these observations, the agent decides an action $A_{ls,t}$ to reschedule the flexible component of $B_t$ to create a modified workload $\hat{B}_t$, thus minimizing the net $CFP = \sum_{t=0}^{N} CFP_t$ over $N$ steps. Here $CFP_t$ will be calculated based on the sum of the DC IT load due to $\hat{B}_t$, the corresponding HVAC cooling load, and the charging and discharging of the battery at every time step.

### A.1.1   Actions ($A_{LS}$)

The action space for $Agent_{LS}$ includes three discrete actions:

---

[4]GitHub repository: https://github.com/HewlettPackard/dc-rl.

[5]Documentation: https://hewlettpackard.github.io/dc-rl.

- *Action 0: Decrease Utilization* - This action attempts to defer the flexible portion of the current workload ($B_{nonflex}$) to a later time. The non-flexible ($B_{flex}$) workload is processed immediately, while the flexible workload is added to a queue for future execution.
- *Action 1: Do Nothing* - This action processes both the flexible ($B_{flex}$) and non-flexible ($B_{nonflex}$) portions of the current workload immediately, without any deferral.
- *Action 2: Increase Utilization* - This action attempts to increase the current utilization by processing tasks from the queue, if available, in addition to the current workload.

### A.1.2 Observations ($S_{LS}$)

The state space observed by the RL agent consists of several features, including:

- **Time of Day** - Represented using sine and cosine transformations of the hour of the day to capture cyclical patterns.
- **Day of the Year** - Represented using sine and cosine transformations to capture seasonal variations.
- **Current Workload** - The current workload level, which includes both flexible and non-flexible components.
- **Queue Status** - The length of the task queue, normalized by the maximum queue length.
- **Grid Carbon Intensity (CI)** - Current and forecasted CI values, capturing the environmental impact of electricity consumption.
- **Battery State of Charge (SoC)** - The current state of charge of the battery, if available.

The observation space is a combination of these features, providing the agent with a comprehensive view of the current state of the environment.

### A.1.3 Mathematical Model

**Workload Breakdown**   Let $B_t$ be the total workload at time $t$. This workload is divided into flexible ($B_{flex,t}$) and non-flexible ($B_{nonflex,t}$) components:

$$B_t = B_{flex,t} + B_{nonflex,t}$$

The flexible workload $B_{flex,t}$ is a fraction of the total workload:

$$B_{flex,t} = \alpha \cdot B_t, \quad 0 < \alpha < 1$$

where $\alpha$ is the flexible workload ratio.

**Actions and Workload Management**   Depending on the action $A_{ls,t}$ chosen by the RL agent, the workload is managed as follows:

1. *Action 0: Decrease Utilization (Queue Flexible Workload)*

$$\hat{B}_t = B_{nonflex,t}$$

The flexible workload $B_{flex,t}$ is added to a task queue $Q_t$ for future execution:

$$Q_{t+1} = Q_t + B_{flex,t}$$

2. *Action 1: Do Nothing*

$$\hat{B}_t = B_t = B_{nonflex,t} + B_{flex,t}$$

There is no change in the task queue:

$$Q_{t+1} = Q_t$$

3. *Action 2: Increase Utilization (Process Queue)*

$$\hat{B}_t = B_t + \min(Q_t, C_{max} - B_t)$$

where $C_{max}$ is the maximum processing capacity. The processed tasks are removed from the task queue:

$$Q_{t+1} = Q_t - \min(Q_t, C_{max} - B_t)$$

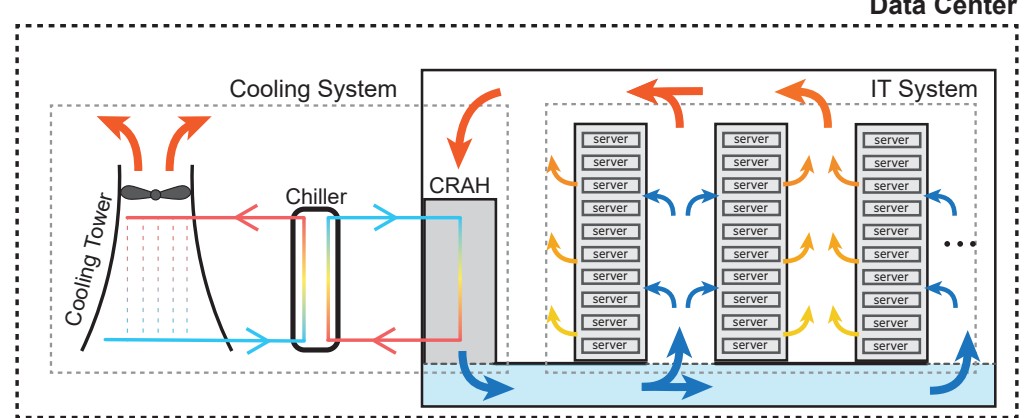

Figure 6: Illustration of the modeled data center, showing the IT section (cabinets and servers) and the Cooling section (Cooling Tower, chiller, and CRAH). The airflow path is also depicted, with cool air supplied through the raised floor and hot air returning via the ceiling. Note: We use CRAH and CRAC interchangeably in the text, but they both represent the same device (CRAH).

## A.2  Data Center Environment ($Env_{DC}$)

The Data Center Environment ($Env_{DC}$) simulates the IT and HVAC operations within a DC, enabling the evaluation of RL algorithms aimed at optimizing cooling setpoints to reduce energy consumption and carbon footprint.

The data center modeled is illustrated in Figure 6. The IT section includes the cabinets and servers, while the Cooling section comprises a Cooling Tower, a chiller, and the Computer Room Air Handler (CRAH). The setup also features a raised floor system that channels cool air from the CRAH to the cabinets. The hot air exits the cabinets and returns to the CRAH via the ceiling.

### A.2.1  Data Center IT Model

Let $\hat{B}_t$ be the net DC workload at time instant $t$ obtained from the Workload Manager. The spatial temperature difference, $\mathbf{\Delta T}_{supply}$, given the DC configuration, is obtained from Computational Fluid Dynamics (CFD). For a given rack, the inlet temperature $T_{inlet,i}$ at $CPU_i$ is computed as:

$$T_{inlet,i,t} = \mathbf{\Delta T}_{supply,i} + T_{CRACsupply,t}$$

where $T_{CRACsupply,t}$ is the CRAC unit supply air temperature. This value is chosen by the RL agent $A_{DC}$.

Next, the CPU power curve $f_{cpu}(inlet\_temp, cpu\_load)$ and IT Fan power curve $f_{itfan}(inlet\_temp, cpu\_load)$ are implemented as linear equations based on (9). Given a server inlet temperature of $T_{inlet,i,t}$ and a processing amount of $\hat{B}_t$ performed by $CPU_i$, the total rack power consumption for rack $k$ across all CPUs from $i = 1$ to $K$, and the total DC Power IT Consumption can be calculated as follows:

$$P_{CPU,t} = \sum_i f_{cpu}(T_{inlet,i,t}, \tilde{B}_t)$$

$$P_{IT\ Fan,t} = \sum_i f_{itfan}(T_{inlet,i,t}, \tilde{B}_t)$$

$$P_{rack,k,t} = P_{CPU,t} + P_{IT\ Fan,t}$$

$$P_{datacenter,t} = \sum_k P_{rack,k,t}$$

### A.2.2 HVAC Cooling Model

Based on the DC IT Load $P_{datacenter,t}$, the IT fan airflow rate, $V_{sfan}$, air thermal capacity $C_{air}$, and air density, $\rho_{air}$, the rack outlet temperature $T_{outlet,i,t}$ is estimated from (9) using:

$$T_{outlet,i,t} = T_{inlet,i,t} + \frac{P_{rack,k,t}}{C_{air} \cdot \rho_{air} \cdot V_{sfan}}$$

In conjunction with the return temperature gradient information $\mathbf{\Delta T}_{return}$ estimated from CFDs, the final CRAC return temperature is obtained as:

$$T_{CRACreturn,t} = \text{avg}(\mathbf{\Delta T}_{return,i} + T_{outlet,i,t})$$

We assume a fixed-speed CRAC Fan unit for circulating air through the IT Room. Hence, the total HVAC cooling load for a given CRAC setpoint $T_{CRACsupply,t}$, return temperature $T_{CRACreturn,t}$, and the mass flow rate $m_{crac,fan}$ is calculated as:

$$P_{cool,t} = m_{crac,fan} \cdot C_{air} \cdot (T_{CRACreturn,t} - T_{CRACsupply,t})$$

To perform $P_{cool,t}$, the amount of cooling, the net chiller load for a chiller with Coefficient of Performance ($COP$) may be estimated as:

$$P_{chiller,t} = P_{cool,t} \left( 1 + \frac{1}{COP} \right)$$

Next, this cooling load is passed on to the cooling tower. Assuming a cooling tower delta as a function of temperature $f_{ct\_delta}(t_{db})$, (21) the required cooling tower air flow rate is calculated as:

$$V_{ct,air,t} = \frac{P_{chiller,t}}{C_{air} \cdot \rho_{air} \cdot f_{ct\_delta}(t_{db})}$$

Finally, the Cooling Tower Load at a flow rate of $V_{ct,air,t}$ is calculated with respect to a reference air flow rate $V_{ct,air,REF}$ and power consumption $P_{ct,REF}$ from the configuration object:

$$P_{CT,t} = P_{ct,REF} \left( \frac{V_{ct,air,t}}{V_{ct,air,REF}} \right)^3$$

Thus, the total HVAC load includes the cooling tower and chiller loads:

$$P_{HVAC,t} = P_{CT,t} + P_{chiller,t}$$

Based on these power values, the IT and HVAC Cooling energy consumptions can be represented as:

$$E_{hvac,t} = P_{HVAC,t} \times \text{step size} \tag{9}$$
$$E_{it,t} = P_{datacenter,t} \times \text{step size} \tag{10}$$

### A.2.3 Actions ($A_{DC}$)

The action space for $Agent_{DC}$ consists of discrete actions representing the adjustment of the CRAC unit's supply air temperature, limited to a range between 16°C to 23°C:

- *Action 0: Decrease Temperature* - The agent decreases the CRAC supply air temperature, enhancing cooling performance but increasing energy consumption.

- *Action 1: Maintain Temperature* - The agent maintains the current CRAC supply air temperature.

- *Action 2: Increase Temperature* - The agent increases the CRAC supply air temperature, which can reduce cooling energy consumption but may increase the IT equipment temperature.

### A.2.4 Observations ($S_{DC}$)

The state space observed by the RL agent consists of several features, including:

- **Time of Day** - Represented using sine and cosine transformations of the hour of the day to capture cyclical patterns.
- **Day of the Year** - Represented using sine and cosine transformations to capture seasonal variations.
- **Ambient Weather** - Includes current temperature and other relevant weather conditions.
- **IT Room Temperature** - Average temperature in the IT room.
- **Energy Consumption** - Previous step cooling and IT energy consumptions.
- **Grid Carbon Intensity (CI)** - Current and forecasted CI values.

The observation space provides a comprehensive view of the current state of the environment to the agent.

### A.2.5 Chiller Sizing

The chiller power consumption is calculated based on the load and operating conditions using the following method:

$$P_{chiller,t} = \text{calculate\_chiller\_power}(max\_cooling\_cap, load, ambient\_temp)$$

**Calculation of Average CRAC Return Temperature**

$$T_{CRACreturn,t} = \text{avg}(\mathbf{\Delta T}_{return,i} + T_{outlet,i,t})$$

**Calculation of HVAC Power**

$$P_{cool,t} = m_{crac,fan} \cdot C_{air} \cdot (T_{CRACreturn,t} - T_{CRACsupply,t})$$

$$P_{chiller,t} = P_{cool,t} \left( 1 + \frac{1}{COP} \right)$$

$$V_{ct,air,t} = \frac{P_{chiller,t}}{C_{air} \cdot \rho_{air} \cdot f_{ct\_delta}(t_{db})}$$

$$P_{CT,t} = P_{ct,REF} \left( \frac{V_{ct,air,t}}{V_{ct,air,REF}} \right)^3$$

$$P_{HVAC,t} = P_{CT,t} + P_{chiller,t}$$

### A.2.6 Water Consumption Model

The water usage for the cooling tower is estimated using a model based on research findings from several key sources. The model accounts for the water loss due to evaporation, drift, and blowdown. The primary references used to develop this model include (22), (23), and guidelines from SPX Cooling Technologies (24).

The water usage model is formulated as follows:

1. **Range Temperature Calculation**: The difference between the hot water temperature entering the cooling tower and the cold water temperature leaving the cooling tower:

$$\text{range\_temp} = \text{hot\_water\_temp} - \text{cold\_water\_temp}$$

where hot_water_temp is the $T_{CRACreturn,t}$, and cold_water_temp is the current CRAC setpoint $T_{CRACsupply,t}$.

2. **Normalized Water Usage**: The baseline water usage per unit time, adjusted for the wet bulb temperature of the ambient air. This accounts for the environmental conditions affecting the cooling tower's efficiency:

$$\text{norm\_water\_usage} = 0.044 \cdot \text{wet\_bulb\_temp} + (0.35 \cdot \text{range\_temp} + 0.1)$$

3. **Total Water Usage**: The normalized water usage is adjusted to ensure non-negativity and further adjusted for drift losses, which are a small percentage of the total water circulated in the cooling tower:

$$\text{water\_usage} = \max(0, \text{norm\_water\_usage}) + \text{norm\_water\_usage} \cdot \text{drift\_rate}$$

4. **Water Usage Conversion**: The total water usage is converted to liters per simulation timestep interval for ease of reporting and consistency with other metrics. Given that we use $N$ timesteps per hour in our simulations, the conversion is as follows:

$$\text{water\_usage\_liters\_per\_timestep} = \left( \frac{\text{water\_usage} \cdot 1000}{N} \right)$$

This model incorporates both theoretical and empirical insights, providing a comprehensive estimation of the water consumption in a data center's cooling tower. By considering the specific operational parameters and environmental conditions, it ensures accurate and reliable water usage calculations, critical for sustainable data center management.

## A.3  Battery Environment ($Env_{BAT}$)

The Battery Environment ($Env_{Bat}$) simulates the battery banks operations within the DC, enabling the evaluation of RL algorithms aimed at optimizing auxiliary battery usage to reduce energy costs and carbon footprint. This environment is a modified version of the battery model from (25).

### A.3.1  Battery Model

The battery model represents the energy storage system, considering its capacity, charging and discharging efficiency, and rate limits. The battery state of charge (SoC) evolves based on the actions taken by the RL agent.

Let $E_{bat,t}$ be the energy stored in the battery at time $t$. The battery can perform three actions: charge, discharge, or remain idle. The maximum battery capacity is $C_{max}$, and the current state of charge is $E_{bat,t}$.

### A.3.2  Actions ($A_{Bat}$)

The action space for $Agent_{Bat}$ includes three discrete actions:

- *Action 0: Charge* - The battery is charged at a rate of $r_{charge}$, consuming $E_{bat,t}$ Wh of energy.

- *Action 1: Idle* - The battery do not consume energy.

- *Action 2: Discharge* - The battery discharges energy at a rate of $r_{discharge}$, supplying $E_{bat,t}$ Wh of energy.

### A.3.3  Observations ($S_{Bat}$)

The state space observed by the RL agent consists of several features, including:

- **Data Center Load** - The current power consumption of the data center.

- **Battery SoC** - The current state of charge of the battery.

- **Grid Carbon Intensity (CI)** - Current and forecasted CI values.

- **Time of Day and Year** - Represented using sine and cosine transformations to capture cyclical patterns.

The observation space is a combination of these features, providing the agent with a comprehensive view of the current state of the environment.

### A.3.4 Mathematical Model

**Battery Charging and Discharging**    The energy stored in the battery evolves based on the action taken:

$$E_{bat,t} = \begin{cases} r_{charge} \cdot \eta_{charge} \cdot \Delta t & \text{if charging} \\ 0 & \text{if idle} \\ r_{discharge} \cdot \eta_{discharge} \cdot \Delta t & \text{if discharging} \end{cases}$$

where $r_{charge}$ and $r_{discharge}$ are the rates of charging and discharging the battery, respectively. These rates determine the amount of energy added to or removed from the battery within a time step $\Delta t$.

**Charging Rate** ($r_{charge}$)    The charging rate $r_{charge}$ is the rate at which energy is added to the battery during the charging process. It is defined as:

$$r_{charge} = \min\left(\frac{C_{max} - E_{bat,t}}{\eta_{charge} \cdot \Delta t}, P_{charge,max}\right)$$

where $P_{charge,max}$ is the maximum allowable charging power. This rate ensures that the battery does not exceed its maximum capacity $C_{max}$ and that charging occurs efficiently.

**Discharging Rate** ($r_{discharge}$)    The discharging rate $r_{discharge}$ is the rate at which energy is drawn from the battery during the discharging process. It is defined as:

$$r_{discharge} = \min\left(\frac{E_{bat,t}}{\eta_{discharge} \cdot \Delta t}, P_{discharge,max}\right)$$

where $P_{discharge,max}$ is the maximum allowable discharging power. This rate ensures that the battery does not discharge below zero and that discharging occurs efficiently.

**Energy Constraints**    The state of charge is bounded by the battery capacity:

$$0 \leq E_{bat,t} \leq C_{max}$$

**Battery Power Constraints**    The maximum power that the battery can charge or discharge is limited by:

$$P_{charge,max} = u \cdot P_{charge} + v$$

$$P_{discharge,max} = u \cdot P_{discharge} + v$$

**Simple Reward Calculation**    The goal of the three agents ($Agent_{LS}$, $Agent_{DC}$, and $Agent_{BAT}$) is to minimize the cumulative carbon footprint (CFP) over a given horizon $N$. The CFP at each time step $t$ is computed as:

$$CFP_t = (E_{it,t} + E_{hvac,t} + E_{bat,t}) \cdot CI_t$$

where:

- $E_{it,t}$: Energy consumption by IT equipment due to $\hat{B}_t$
- $E_{hvac,t}$: Energy consumption by HVAC systems
- $E_{bat,t}$: Energy contribution from the battery (positive when discharging, negative when charging)
- $CI_t$: Grid carbon intensity at time $t$

The total reward is then:

$$R = -\sum_{t=0}^{N} CFP_t$$

The reward could have other terms that may consider queue length, water usage, average task delay, etc.

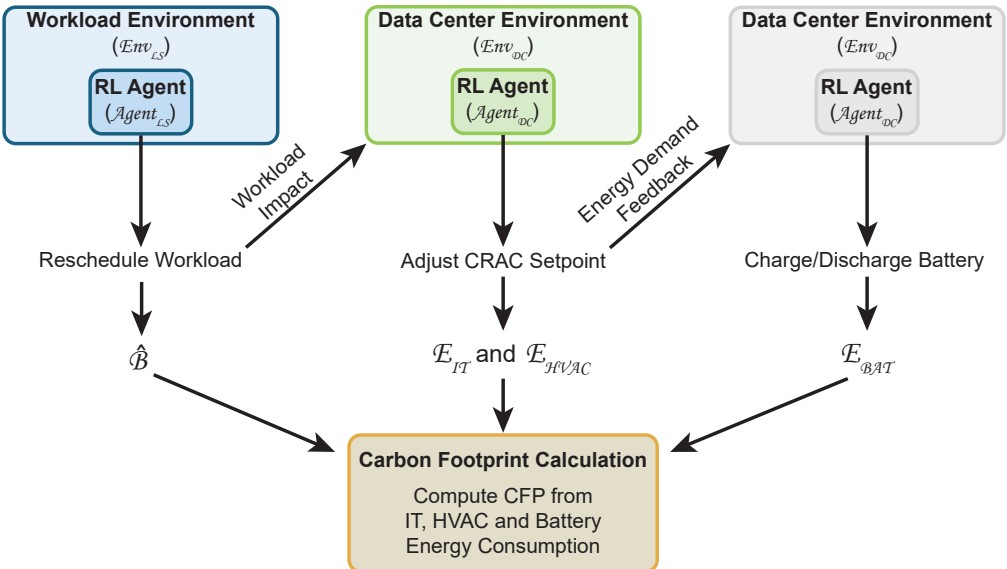

Figure 7: Interconnection of environments and agent actions. The figure shows how the Workload Environment ($Env_{LS}$) interacts with the Data Center Environment ($Env_{DC}$) by rescheduling workloads, and how the Data Center Environment impacts the Battery Environment ($Env_{BAT}$) through energy demands. Each agent observes the state of its respective environment and takes actions to optimize operations, with the overall goal of minimizing the carbon footprint (CFP) through coordinated efforts.

## A.4 Interconnection of Environments and Agent Actions

Figure 7 illustrates the interconnection of the different environments ($Env_{LS}$, $Env_{DC}$, and $Env_{BAT}$) and the actions of their respective RL agents. This diagram highlights how the decisions made by each agent impact the overall DC operations and contribute to the optimization of energy consumption and carbon footprint.

In the **Workload Environment** ($Env_{LS}$), the RL agent ($Agent_{LS}$) reschedules flexible workloads to optimize utilization. This action will influence the IT load, which directly impacts the **Data Center Environment** ($Env_{DC}$). The RL agent ($Agent_{DC}$) in the data center environment adjusts the CRAC setpoints to optimize cooling and IT operations, thus affecting the HVAC cooling load and overall energy consumption.

The **Battery Environment** ($Env_{BAT}$) is influenced by the energy demands of the data center environment. The RL agent ($Agent_{BAT}$) manages the charging and discharging of the battery to optimize energy usage and reduce the carbon footprint. The interconnections between these environments ensure that the agents work together to minimize the cumulative CFP by considering the energy consumption of IT, HVAC, and battery systems.

By observing the current state and forecast data, each agent makes informed decisions that contribute to the overall sustainability and efficiency of the data center operations. This coordinated approach leverages the strengths of each environment to achieve significant reductions in energy consumption and carbon emissions.

## B   Customization of *dc_config.json*

The customization of the DC is done through the `dc_config.json` file located in the utils folder. This file allows users to specify every aspect of the DC environment design. We show here a part of the configuration file to indicate the different configurable elements inside SustainDC. Additional elements can be added to this config either under an existing section or a new section, and `utils/dc_config_reader.py` will automatically import the new configurations. Inside the `"data_center_configuration"` SustainDC allows the user to configure the dimensions of the data

center arrangement, the compiled CFD supply and approach temperature delta values and the maximum allowable CPUs per rack. There is an extensive set of parameters that can be configured under the "hvac_configuration" section including physical constants, parameters of the computer room air-conditioning unit (CRAC), chiller, pumps and cooling towers. The "server_characteristics" block allows the user to specify the properties of individual servers in the data center, including their idle power, full load fan frequency and power.

```
{
"data_center_configuration" :
{
    "NUM_ROWS" : 4,
    "NUM_RACKS_PER_ROW" : 5,
    "RACK_SUPPLY_APPROACH_TEMP_LIST" : [
                                        5.3, 5.3, 5.3, 5.3,5.3,
                                        5.0, 5.0, 5.0, 5.0,5.0,
                                        5.0, 5.0, 5.0, 5.0,5.0,
                                        5.3, 5.3, 5.3, 5.3, 5.3
                                        ],
    "RACK_RETURN_APPROACH_TEMP_LIST" :  [
                                        -3.7, -3.7, -3.7, -3.7, -3.7,
                                        -2.5, -2.5, -2.5, -2.5, -2.5,
                                        -2.5, -2.5, -2.5, -2.5, -2.5,
                                        -3.7, -3.7, -3.7, -3.7, -3.7
                                        ],
    "CPUS_PER_RACK" : 200
},
"hvac_configuration" :
{
    "C_AIR" : 1006,
    "RHO_AIR" : 1.225,
    "CRAC_SUPPLY_AIR_FLOW_RATE_pu" : 0.00005663,
    "CRAC_REFRENCE_AIR_FLOW_RATE_pu" : 0.00009438,
    "CRAC_FAN_REF_P" : 150,
    "CHILLER_COP_BASE" : 5.0,
    "CHILLER_COP_K" : 0.1,
    "CHILLER_COP_T_NOMINAL" : 25.0,
    "CT_FAN_REF_P" : 1000,
    "CT_REFRENCE_AIR_FLOW_RATE" : 2.8315,
    "CW_PRESSURE_DROP" : 300000,
    "CW_WATER_FLOW_RATE" : 0.0011,
    "CW_PUMP_EFFICIENCY" : 0.87,
    "CT_PRESSURE_DROP" : 300000,
    "CT_WATER_FLOW_RATE" : 0.0011,
    "CT_PUMP_EFFICIENCY" : 0.87

},
"server_characteristics" :
{
    "CPU_POWER_RATIO_LB" : [0.01, 1.00],
    "CPU_POWER_RATIO_UB" : [0.03, 1.02],
    "IT_FAN_AIRFLOW_RATIO_LB" : [0.01, 0.225],
    "IT_FAN_AIRFLOW_RATIO_UB" : [0.225, 1.0],
    "IT_FAN_FULL_LOAD_V" : 0.051,
    "ITFAN_REF_V_RATIO" : 1.0,
    "ITFAN_REF_P" : 10.0,
    "INLET_TEMP_RANGE" : [16, 28],
    "DEFAULT_SERVER_POWER_CHARACTERISTICS":[[170, 20],
                                            [120, 10],
                                            [130, 10],
                                            [130, 10],
                                            [130, 10],
                                            [130, 10],
                                            [130, 10],
                                            [130, 10],
                                            [130, 10],
                                            [130, 10],
                                            [130, 10],
                                            [130, 10],
                                            [130, 10],
                                            [170, 10],
                                            [130, 10],
                                            [130, 10],
                                            [110, 10],
                                            [170, 10],
                                            [170, 10],
                                            [170, 10]],
    "HP_PROLIANT" : [110,170]
  }
}
```

# C  Performance of RL agents on Evaluation Metrics

In this section, we provide the numerical results we obtained from the main paper. The results are shown in Tables 2 (advantage of multiagent vs single agent), 3 (effects of reward sharing across agents), 4, 5, 6 and 7 (ablation across geographical locations with different weather, grid carbon intensity and server load pattern). We observed that there is not a single algorithm that works well across different metrics and geographical locations, and this is visually appreciated in the main paper.

Table 2: Performance with respect to evaluation metrics on single and multiple RL agent baselines.
$A_* : RL\ agent\ B_* : non-RL\ baseline\ agent$

| Evaluation Metric → Algorithm ↓ | $CFP$ (kgCO2) | HVAC Energy (kwh) | IT Energy (kwh) | Task Queue | Water Usage (litre) |
|---|---|---|---|---|---|
| 1:$A_{LS} + B_{DC} + B_{BAT}$ | 167.61 | 391.6 | 1033.8 | 0.52 | 10433.46 |
| 2:$B_{LS} + A_{DC} + B_{BAT}$ | 153.56 | 372.9 | 944.5 | 0.0 | 10930.77 |
| 3:$B_{LS} + B_{DC} + A_{BAT}$ | 168.22 | 390.3 | 1029.8 | 0.0 | 10493.95 |
| 4: $A_{LS} + A_{DC} + B_{BAT}$ | 155.97 | 374.9 | 941.3 | 0.48 | 10883.73 |
| 5:$A_{LS} + B_{DC} + A_{BAT}$ | 168.64 | 391.1 | 1030.9 | 0.56 | 10470.43 |
| 6:$B_{LS} + A_{DC} + A_{BAT}$ | 155.44 | 374.8 | 942.5 | 0 | 10883.73 |
| 7:$A_{LS} + A_{DC} + A_{BAT}$ | 155.23 | 371.8 | 937.4 | 0.45 | 10826.61 |

Table 3: IPPO evaluated on SustainDC with different values of collaborative reward coefficient $\alpha$ (Average result over 12 runs)

| Evaluation Metric → Algorithm ↓ | $CFP$ (kgCO2) | HVAC Energy (kwh) | IT Energy (kwh) | Task Queue | Water Usage (litre) |
|---|---|---|---|---|---|
| IPPO($\alpha = 1.0$) | 176.3 | 415.2 | 932.8 | 12.5 | 445.6 |
| IPPO($\alpha = 0.8$) | 176.2 | 414.6 | 932.8 | 9.5 | 445.8 |
| IPPO($\alpha = 0.1$) | 176.4 | 415.3 | 932.9 | 15.7 | 446.2 |

Table 4: Multiagent RL framework evaluated on SustainDC for a data center located in New York (Average result over 5 runs)

| Evaluation Metric → Algorithm ↓ | $CFP$ (kgCO2) | HVAC Energy (kwh) | IT Energy (kwh) | Task Queue | Water Usage (litre) |
|---|---|---|---|---|---|
| IPPO | 179.6 | 417.1 | 945.9 | 20.9 | 446.2 |
| MAPPO | 176.4 | 417.0 | 932.7 | 19.6 | 446.2 |
| HAPPO | 177.3 | 414.8 | 930.9 | 12.8 | 441.9 |
| HAA2C | 177.5 | 419.0 | 934.8 | 25.2 | 14977.1 |
| HAD3QN | 178.4 | 420.5 | 940.4 | 28.0 | 14950.9 |
| HASAC | 181.7 | 424.2 | 960.8 | 79.7 | 14842.4 |

# D  Agents/Env behavior

## D.1  Battery

The battery environment demonstrates how the battery's state of charge (SoC) and actions evolve over time under random behaviors. These figures illustrate two different examples generated using distinct random seeds.

Figure 8 shows the battery's SoC and the actions taken (Charge, Discharge, Idle) over simulated days for two different random behaviors.

Table 5: Multiagent RL framework evaluated on SustainDC for a data center located in Georgia (Average result over 5 runs)

| Evaluation Metric → Algorithm ↓ | $CFP$ (kgCO2) | HVAC Energy (kwh) | IT Energy (kwh) | Task Queue | Water Usage (litre) |
|---|---|---|---|---|---|
| IPPO | 265.4 | 376.7 | 935.4 | 6.8 | 31773.5 |
| MAPPO | 263.4 | 370.3 | 935.9 | 0.35 | 31949.9 |
| HAPPO | 264.1 | 370.4 | 929.0 | 0.47 | 31890.7 |
| HAA2C | 262.7 | 367.1 | 928.3 | 6.6 | 32071.5 |
| HAD3QN | 262.8 | 370.7 | 935.1 | 0.0 | 31952.2 |
| HASAC | 263.0 | 367.4 | 932.4 | 0.0 | 32135.7 |

Table 6: Multiagent RL framework evaluated on SustainDC for a data center located in California (Average result over 5 runs)

| Evaluation Metric → Algorithm ↓ | $CFP$ (kgCO2) | HVAC Energy (kwh) | IT Energy (kwh) | Task Queue | Water Usage (litre) |
|---|---|---|---|---|---|
| IPPO | 170.0 | 384.3 | 933.8 | 12.9 | 28141.4 |
| MAPPO | 159.3 | 388.2 | 936.1 | 19.5 | 33289.3 |
| HAPPO | 159.1 | 376.3 | 935.8 | 74.9 | 30141.8 |
| HAA2C | 158.7 | 381.7 | 933.5 | 54.1 | 30135.4 |
| HAD3QN | 161.5 | 378.4 | 929.6 | 25.8 | 30017.4 |
| HASAC | 172.9 | 434.4 | 1027.0 | 43.8 | 29277.5 |

Figure 9 compares the energy consumption with and without the battery over simulated days for two different random behaviors. This comparison illustrates the impact of battery usage on the overall energy consumption of the data center.

Figure 10 shows the energy added to and removed from the battery over simulated days for two different random behaviors. These figures demonstrate how the battery charges and discharges energy, providing insights into its operational patterns.

# E   External variables

## E.1   Workload

The *Workload* external variable in SustainDC represents the computational demand placed on the data center. Workload traces are provided in the form of FLOPs (floating-point operations) required by various jobs. By default, SustainDC includes a collection of open-source workload traces from *Alibaba* (10) and *Google* (11) data centers. Users can customize this component by

Table 7: Multiagent RL framework evaluated on SustainDC for a data center located in Arizona (Average result over 5 runs)

| Evaluation Metric → Algorithm ↓ | $CFP$ (kgCO2) | HVAC Energy (kwh) | IT Energy (kwh) | Task Queue | Water Usage (litre) |
|---|---|---|---|---|---|
| IPPO | 408.7 | 380.8 | 934.8 | 0.60 | 30251.6 |
| MAPPO | 410.8 | 383.3 | 947.5 | 502.4 | 31289.6 |
| HAPPO | 405.5 | 381.9 | 936.6 | 0.26 | 30983.7 |
| HAA2C | 407.1 | 385.0 | 929.9 | 7.54 | 32706.3 |
| HAD3QN | 405.6 | 386.4 | 1094.0 | 0.0051 | 30377.3 |
| HASAC | 404.6 | 380.8 | 936.7 | 0.54 | 30878.7 |

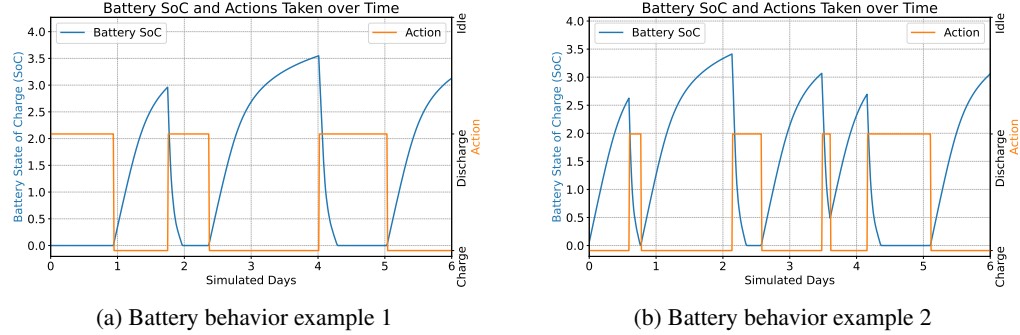

(a) Battery behavior example 1           (b) Battery behavior example 2

Figure 8: Battery State of Charge (SoC) and actions taken over time under two different random behaviors. The actions are labeled as Charge, Discharge, and Idle.

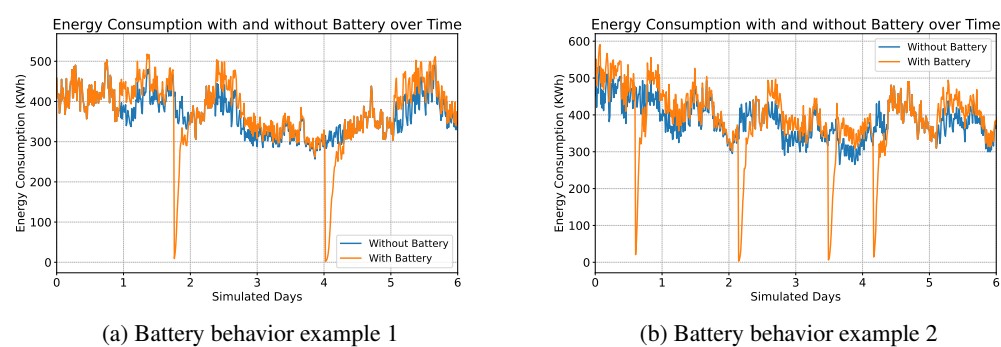

(a) Battery behavior example 1           (b) Battery behavior example 2

Figure 9: Energy consumption with and without the battery over time under two different random behaviors. The comparison illustrates the effect of battery usage on overall energy consumption.

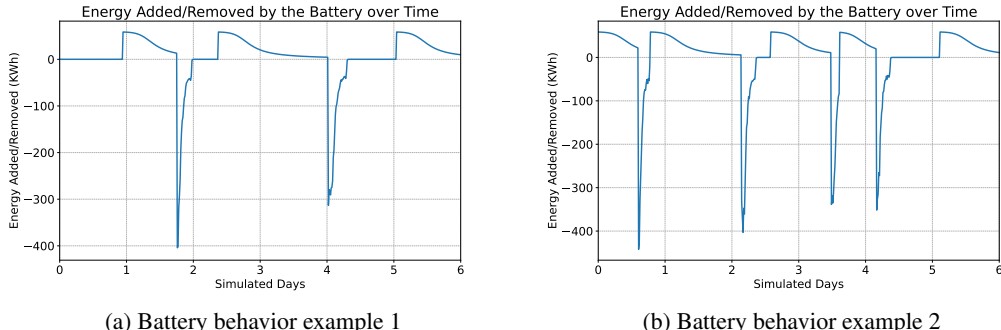

(a) Battery behavior example 1           (b) Battery behavior example 2

Figure 10: Energy added to and removed from the battery over time under two different random behaviors. The figures show how the battery charges and discharges energy throughout the simulated period.

adding new workload traces to the *data/Workload* folder or specifying a path to existing traces in the *sustaindc_env.py* file under the `workload_file` configuration. Below is an example of modifying the workload configuration:

```
class EnvConfig(dict):

    DEFAULT_CONFIG = {
        "workload_file": "data/Workload/Alibaba_CPU_Data_Hourly_1.csv",
        ...
    }
```

The workload file should contain one year of data with an hourly periodicity (365*24=8760 rows). The file structure should have two columns, where the first column does not have a name, and the second column should be named `cpu_load`. Below is an example of the file structure:

```
,cpu_load
1,0.380
2,0.434
3,0.402
4,0.485
...
```

Figure 11 shows examples of different workload traces from Alibaba (v2017) and Google (v2011) data centers. Figure 12 provides a comparison between two workload traces of Alibaba (v2017) and Google (v2011).

## E.2 Weather

The *Weather* external variable in SustainDC captures the ambient environmental conditions impacting the data center's cooling requirements. By default, SustainDC includes weather data files in the .epw format from `https://energyplus.net/weather` for various locations where data centers are commonly situated. These locations include Arizona, California, Georgia, Illinois, New York, Texas, Virginia, and Washington. Users can customize this component by adding new weather files to the *data/Weather* folder or specifying a path to existing weather files in the *sustaindc_env.py* file under the `weather_file` configuration. Below is an example of modifying the weather configuration:

```
class EnvConfig(dict):

    DEFAULT_CONFIG = {
        'weather_file': 'data/Weather/USA_NY_New.York-Kennedy.epw',
        ...
    }
```

Each .epw file contains hourly data for various weather parameters, but for our purposes, we focus on the ambient temperature. Figure 13 shows the typical average ambient temperature across different locations over one year. Figure 14 provides a comparison of external temperatures across the different selected locations.

## E.3 Carbon Intensity

The *Carbon Intensity (CI)* external variable in SustainDC represents the carbon emissions associated with electricity consumption. By default, SustainDC includes CI data files for various locations:

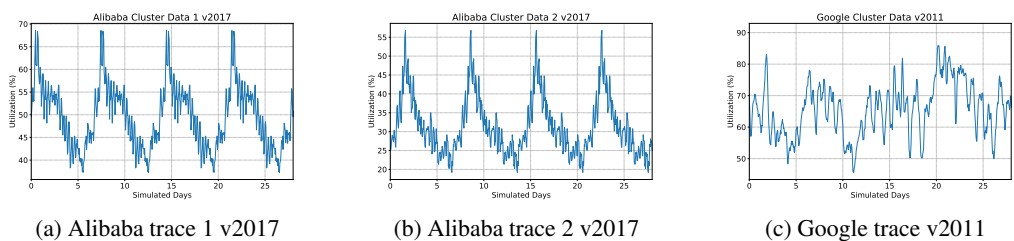

(a) Alibaba trace 1 v2017      (b) Alibaba trace 2 v2017      (c) Google trace v2011

Figure 11: Examples of different workload traces from Alibaba and Google data centers.

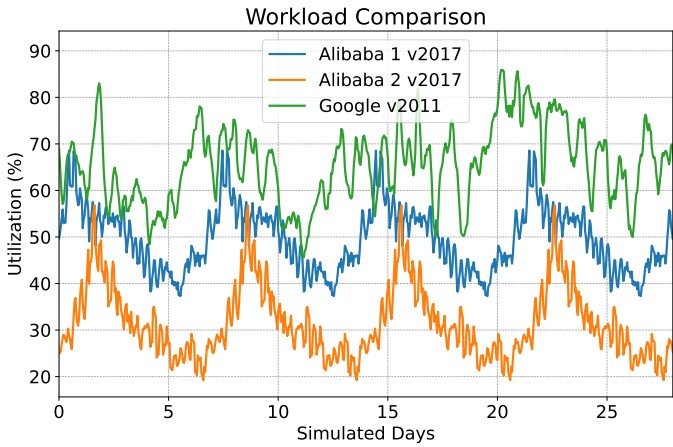

Figure 12: Comparison between two workload traces of Alibaba trace (2017) and Google (2011).

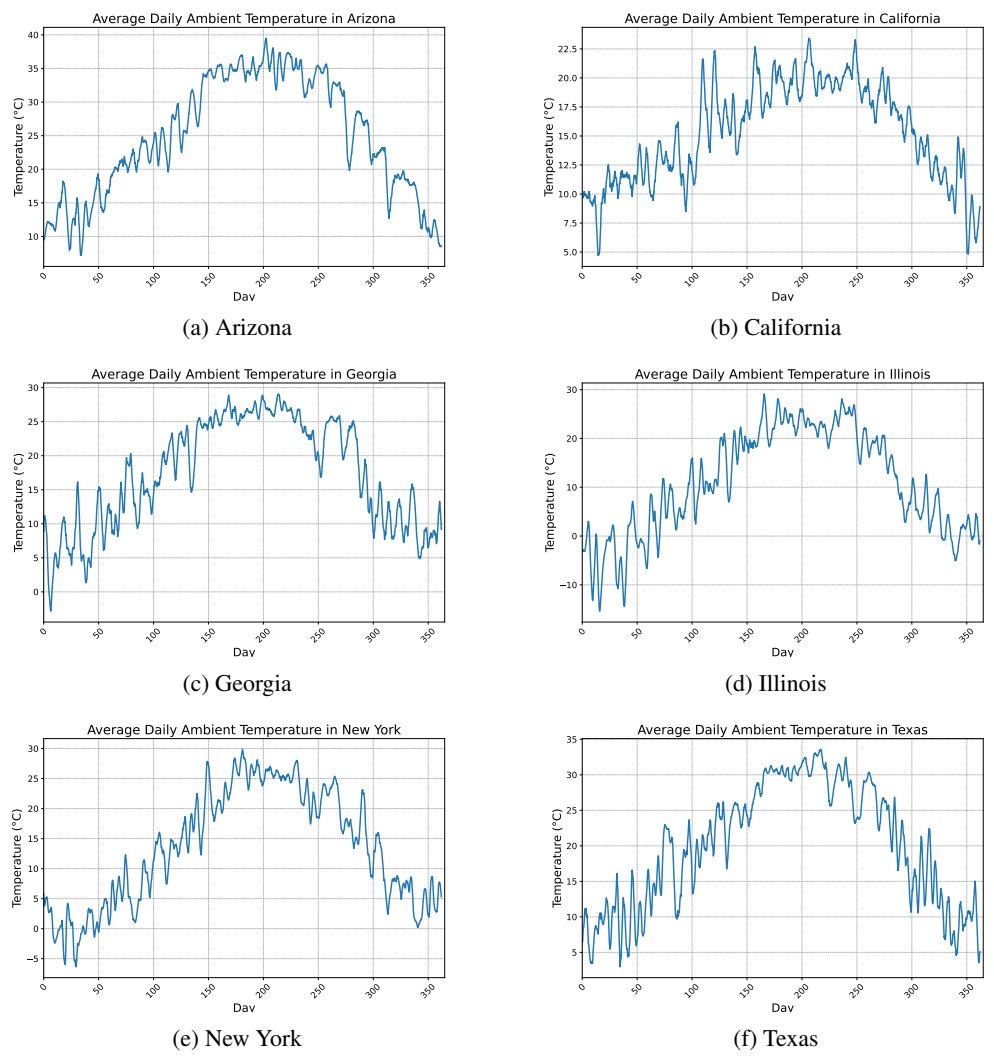

(a) Arizona

(b) California

(c) Georgia

(d) Illinois

(e) New York

(f) Texas

Figure 13: Typical average ambient temperature across different locations across one year.

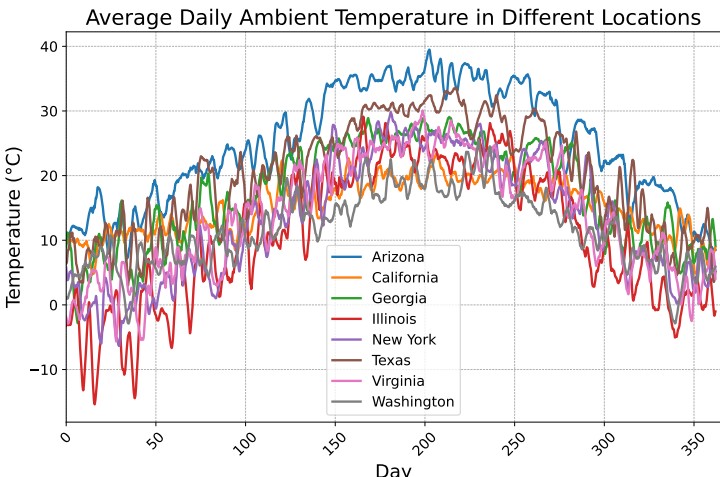

Figure 14: Comparison between external temperature of the different selected locations.

Arizona, California, Georgia, Illinois, New York, Texas, Virginia, and Washington. These files are located in the *data/CarbonIntensity* folder and are extracted from `https://api.eia.gov/bulk/EBA.zip`. Users can customize this component by adding new CI files to the *data/CarbonIntensity* folder or specifying a path to existing files in the *sustaindc_env.py* file under the `cintensity_file` configuration. Below is an example of modifying the CI configuration:

```
class EnvConfig(dict):

    DEFAULT_CONFIG = {
        'cintensity_file': 'data/CarbonIntensity/NY_NG_&_avgCI.csv',
        ...
    }
```

The CI file should contain one year of data with an hourly periodicity (365*24=8760 rows). The file structure should have the following columns: `timestamp`, `WND`, `SUN`, `WAT`, `OIL`, `NG`, `COL`, `NUC`, `OTH`, and `avg_CI`. `WND`, `SUN`, `WAT`, `OIL`, `NG`, `COL`, `NUC`, and `OTH` represent the energy sources contributing to the carbon intensity. These sources include wind, solar, water, oil, natural gas, coal, nuclear, and other types of energy, respectively. Below is an example of the file structure:

```
timestamp,WND,SUN,WAT,OIL,NG,COL,NUC,OTH,avg_CI
2022-01-01 00:00:00+00:00,1251,0,3209,0,15117,2365,4992,337,367.450
2022-01-01 01:00:00+00:00,1270,0,3022,0,15035,2013,4993,311,363.434
2022-01-01 02:00:00+00:00,1315,0,2636,0,14304,2129,4990,312,367.225
2022-01-01 03:00:00+00:00,1349,0,2325,0,13840,2334,4986,320,373.228
...
```

In Figure 15, the average daily carbon intensity for each selected location is shown, highlighting the variations in carbon emissions associated with electricity consumption across different regions.

In Figure 16, a comparison of carbon intensity across all the selected locations is presented, providing a comprehensive overview of how carbon emissions vary between these areas.

In Figure 17, we show the average daily carbon intensity against the average daily coefficient of variation (CV) for various locations. This figure highlights an important perspective on the variability and magnitude of carbon intensity values across different regions. Locations with a high CV indicate greater fluctuation in carbon intensity, offering more "room to play" for DRL agents to effectively reduce carbon emissions through dynamic actions. Additionally, locations with a high average carbon intensity value present greater opportunities for achieving significant carbon emission reductions. The selected locations are highlighted, while other U.S. locations are also plotted for comparison. Regions with both high CV and high average carbon intensity are identified as prime targets for DRL agents to maximize their impact on reducing carbon emissions.

In the table bellow (8) is the summarizing the selected locations, typical weather values, and carbon emissions characteristics:

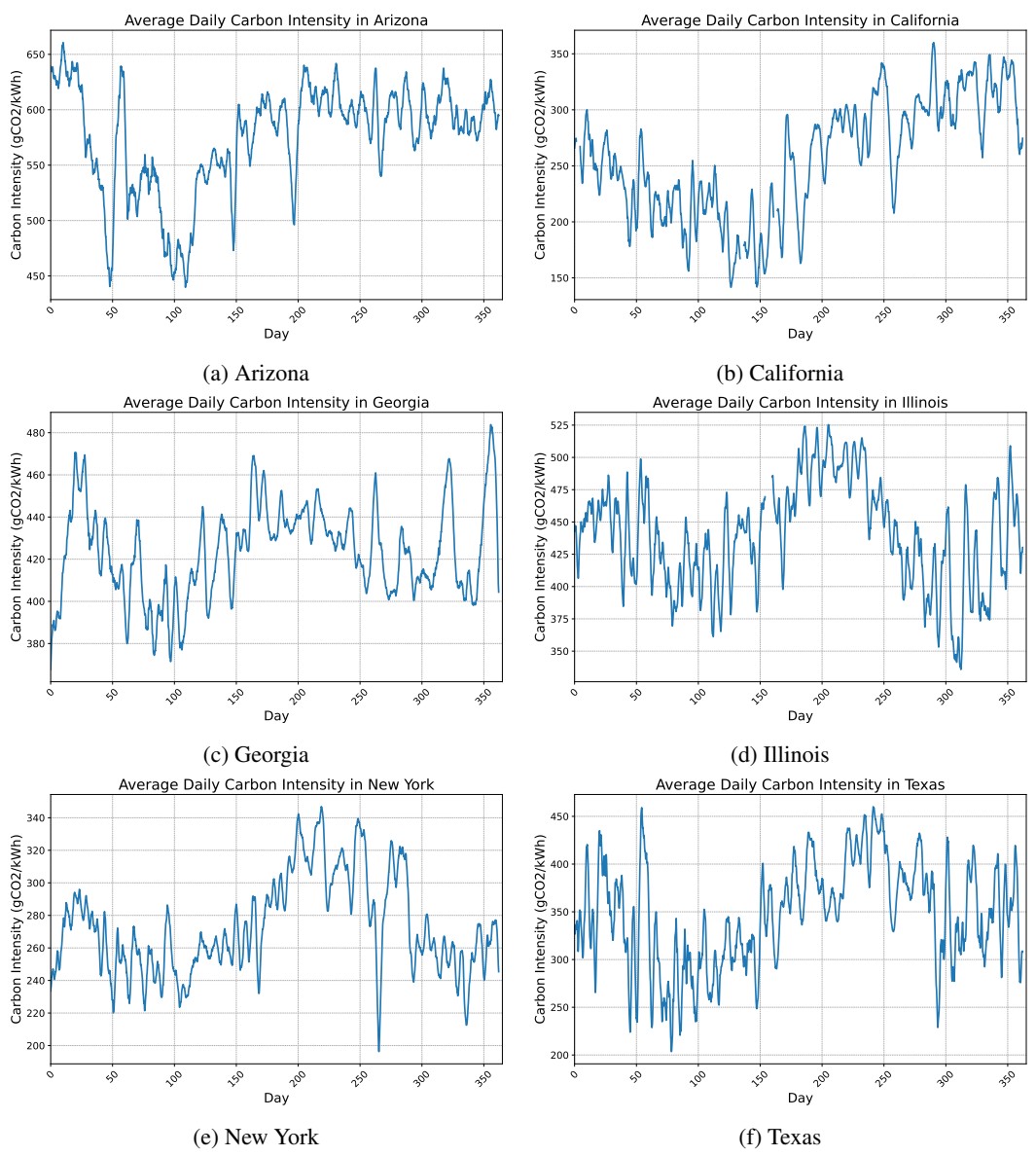

Figure 15: Typical average carbon intensity across different locations over one year.

| Location | Typical Weather | Carbon Emissions |
|---|---|---|
| Arizona | Hot, dry summers; mild winters | High avg CI, High variation |
| California | Mild, Mediterranean climate | Medium avg CI, Medium variation |
| Georgia | Hot, humid summers; mild winters | High avg CI, Medium variation |
| Illinois | Cold winters; hot, humid summers | High avg CI, Medium variation |
| New York | Cold winters; hot, humid summers | Medium avg CI, Medium variation |
| Texas | Hot summers; mild winters | Medium avg CI, High variation |
| Virginia | Mild climate, seasonal variations | Medium avg CI, Medium variation |
| Washington | Mild, temperate climate; wet winters | Low avg CI, Low variation |

Table 8: Summary of Selected Locations with Typical Weather and Carbon Emissions Characteristics

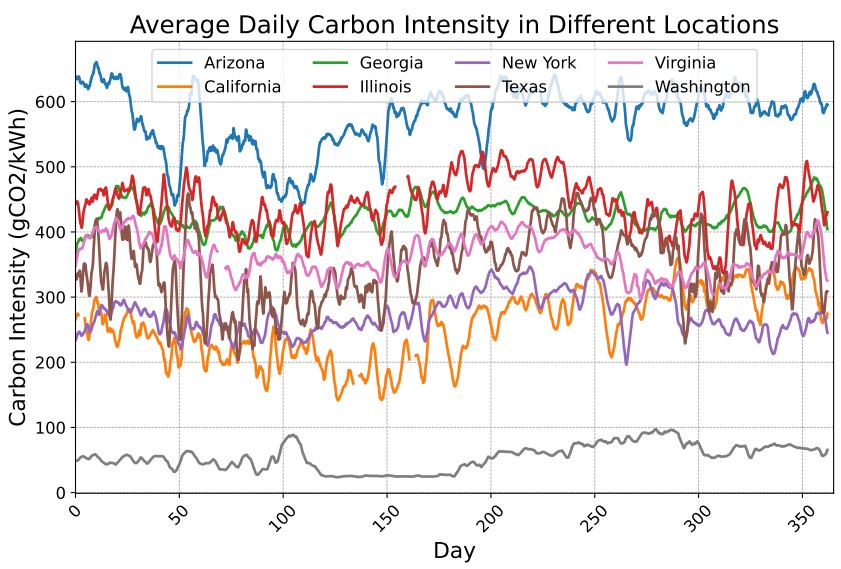

Figure 16: Comparison of carbon intensity across the different selected locations.

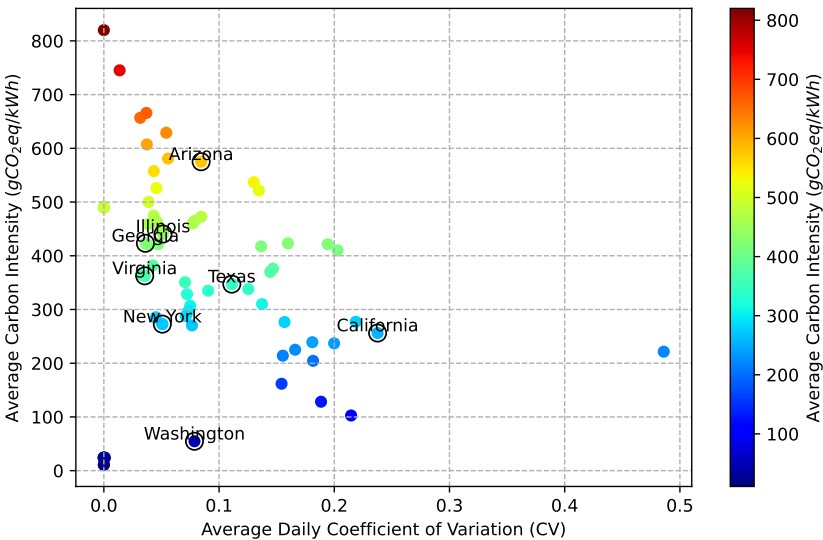

Figure 17: Average daily carbon intensity versus average daily coefficient of variation (CV) for the grid energy provided from US. Selected locations are remarked. High CV indicates more fluctuation, providing more opportunities for DRL agents to reduce carbon emissions. High average carbon intensity values offer greater potential gains for DRL agents.

| State | Data Centers |
|---|---|
| California | 254 |
| Virginia | 250 |
| Texas | 239 |
| New York | 128 |
| Illinois | 122 |
| Florida | 120 |
| Ohio | 98 |
| Washington | 84 |
| Georgia | 75 |
| New Jersey | 69 |

Table 9: Summary of U.S. States with the Most Data Centers (ref: (26))

Considering the data from (26), the U.S. states with the highest number of data centers are summarized in Table 9. The states with the most significant number of data centers tend to be Virginia, Texas, California, and New York. Virginia, especially, is a major hub due to its proximity to Washington D.C. and the abundance of fiber optic cable networks. Texas and California are also prominent due to their size, economic output, and significant tech industries. New York, particularly around New York City, hosts numerous data centers that serve the financial sector and other industries.

The selection of these locations is justified by their significant number of data centers, which emphasizes the potential impact of DRL agents in these regions. By targeting areas with both high data center density and favorable carbon intensity characteristics, DRL agents can maximize their effectiveness in reducing carbon emissions.

# F    Reward Evaluation and Customization

## F.1    Load Shifting Penalty ($LS_{Penalty}$)

The Load Shifting Penalty ($LS_{Penalty}$) is applied to the Load Shifting Agent ($Agent_{LS}$) in the Workload Environment ($Env_{LS}$) if it fails to reschedule flexible workloads within the same day. If $D_t$ (the amount of rescheduled workload left) is positive at the end of the day, $penalty\_tasks\_queue$ is assigned. Additionally, we included a function that progressively increases the penalty as the hour of the day approaches 24h. This means the penalty increases linearly from hour 23h to hour 24h.

Furthermore, there is a penalty for tasks that were dropped due to queue limits ($penalty\_dropped\_tasks$). This penalty is added to discourage the agent from dropping tasks and ensure that workloads are managed efficiently.

Therefore, the $LS_{Penalty}$ is composed of $penalty\_tasks\_queue$ and $penalty\_dropped\_tasks$. Related work in this area include (27; 28; 29; 30; 31; 32; 33).

## F.2    Default Reward Function

The default reward function used in SustainDC for the Load Shifting Agent is implemented as follows:

```
def default_ls_reward(params: dict) -> float:
    """
    Calculate the reward value based on normalized load shifting
    and energy consumption.

    Parameters:
        params (dict): Dictionary containing parameters:
            - bat_total_energy_with_battery_KWh (float):
                Total energy consumption with battery.
            - norm_CI (float): Normalized carbon intensity.
            - bat_dcload_min (float): Minimum data center load.
```

```
        - bat_dcload_max (float): Maximum data center load.
        - ls_tasks_dropped (int): Number of tasks dropped due to queue limit.
        - ls_tasks_in_queue (int): Number of tasks currently in queue.
        - ls_current_hour (int): Current hour in the simulation.

    Returns:
        float: Calculated reward value.
    """
    # Energy part of the reward
    total_energy_with_battery = params['bat_total_energy_with_battery_KWh']
    norm_CI = params['norm_CI']
    dcload_min = params['bat_dcload_min']
    dcload_max = params['bat_dcload_max']

    # Calculate the reward associated with the energy consumption
    norm_net_dc_load = (total_energy_with_battery - dcload_min) /
                        (dcload_max - dcload_min)
    footprint = -1.0 * norm_CI * norm_net_dc_load

    # Penalize the agent for each task that was dropped due to queue limit
    penalty_per_dropped_task = -10  # Define the penalty value per dropped task
    tasks_dropped = params['ls_tasks_dropped']
    penalty_dropped_tasks = tasks_dropped * penalty_per_dropped_task

    tasks_in_queue = params['ls_tasks_in_queue']
    current_step = params['ls_current_hour']
    penalty_tasks_queue = 0

    if current_step % (24*4) >= (23*4):  # Penalty for queued tasks at the
                                         #          end of the day
        factor_hour = (current_step % (24*4)) / 96  # min = 0.95833, max = 0.98953
        factor_hour = (factor_hour - 0.95833) / (0.98935 - 0.95833)
        penalty_tasks_queue = -1.0 * factor_hour * tasks_in_queue / 10  # Penalty
                                                         for each task left in the queue

    LS_penalty = penalty_dropped_tasks + penalty_tasks_queue

    reward = footprint + LS_penalty

    return reward
```

## F.3   Customization of Reward Formulations

Users can choose to use any other reward formulation by defining custom reward functions inside
*utils/reward_creator.py*. To create a custom reward function, you can define it as follows:

```
def custom_reward(params: dict) -> float:
    # Custom reward calculation logic
    pass
```

Replace the logic inside the `custom_reward` function with your custom reward logic.

For more examples of custom reward functions, users can check the file *utils/reward_creator.py*.

To use the custom reward function, you need to include it in the *utils/reward_creator.py* as follows:

```
# Other reward methods can be added here.

REWARD_METHOD_MAP = {
    'default_dc_reward' : default_dc_reward,
    'default_bat_reward': default_bat_reward,
    'default_ls_reward' : default_ls_reward,
    # Add custom reward methods here
    'custom_reward' : custom_reward,
}
```

Additionally, you need to specify the reward function in *harl/configs/envs_cfgs/dcrl.yaml*:

```
agents:
...
ls_reward: default_ls_reward
dc_reward: default_dc_reward
bat_reward: default_bat_reward
...
```

This flexibility ensures that SustainDC can be adapted to a wide range of research and operational needs in sustainable data center management.

