# SustainDC: Benchmarking for Sustainable Data Center Control
## Supplementary Information

**Avisek Naug**[†]**, Antonio Guillen**[†]**, Ricardo Luna**[†]**, Vineet Gundecha**[†]**, Cullen Bash,
Sahand Ghorbanpour**, **Sajad Mousavi**, **Ashwin Ramesh Babu**, **Dejan Markovikj**,
**Lekhapriya D Kashyap**, **Desik Rengarajan**, **Soumyendu Sarkar**[†][*]

Hewlett Packard Enterprise (Hewlett Packard Labs)

{avisek.naug, antonio.guillen, rluna, vineet.gundecha, cullen.bash, sahand.ghorbanpour,
sajad.mousavi, ashwin.ramesh-babu, dejan.markovikj, lekhapriya.dheeraj-kashyap, desik.rengarajan,
soumyendu.sarkar}@hpe.com

## Contents

[*]Corresponding author. [†]These authors contributed equally.

38th Conference on Neural Information Processing Systems (NeurIPS 2024) Track on Datasets and Benchmarks.