# OpenReview forum: "SustainDC: Benchmarking for Sustainable Data Center Control"
_NeurIPS.cc/2024/Datasets_and_Benchmarks_Track — NeurIPS 2024 Track Datasets and Benchmarks Poster_

### Official Review · Reviewer_xaCY · 2024-06-25
**Interesting work, useful for improving Data Center operations while reducing energy and carbon emissions**

**Rating:** 7
**Confidence:** 4
**Correctness:** The claims sound correct
**Clarity:** The paper is well-structured and clear.

**Review:**

A good quality paper that is clearly written and has substantial results for publishing
Table 1 also gives a high-level understanding of the control policies in the environment.
Fig 4 has interesting results  on understanding the efficiency of DC operations in various regions along various dimensions

**Strengths:**

The environment is user-friendly and offers customization options compared to the state-of-the-art methods.

**Additional Feedback:**

N/A

**Documentation:**

N/A - Benchmarking suite

**Limitations:**

Limitations are adequately addressed

**Opportunities For Improvement:**

It would be interesting to add experiments to see  energy savings for a typical day with highly varying workload and constant workload, to evaluate the robustness of the python environment against other simulation suites.

**Relation To Prior Work:**

Clearly talked about the state-of -the-art

**Summary And Contributions:**

Provides a Python suite for benchmarking multi-agent reinforcement learning algorithms for data centers

---

> ### Author Rebuttal · Authors · 2024-08-16
>
> We thank reviewer xaCY for the constructive feedback. Please see our response below.
>
> "It would be interesting to add experiments to see energy savings for a typical day with highly varying workload and constant workload, to evaluate the robustness of the python environment against other simulation suites.: "
>
> We will run additional experiments on the recommended workload patterns with different levels of variability and evaluate the robustness of the simulation environment. We plan to add a benchmarking with respect to the EnergyPlus and Modelica. Currently, we compare the speed of simulation for SustainDC vs EnergyPlus for a data center with same number of server in the figure attached in the PDF. We will add further information on this topic to the open-source repository and documentation.

---

### Official Review · Reviewer_XQ3a · 2024-07-19
**Valuable work, but the new implementation requires validation against EnergyPlus or Modelica**

**Rating:** 6
**Confidence:** 4
**Correctness:** Yes. The code is also easy to run, re…
**Clarity:** Yes. The documentation and supplement…

**Review:**

This work presents a valuable contribution to the field of data center efficiency and reinforcement learning. The authors have developed a well-implemented, easily accessible platform for studying data center cooling optimization using RL techniques. However, I think validation against EnergyPlus or Modelica is critical. Also, additional features like random/rule-based baselines and carbon/weather data generators would complete the package.

Pros:
* Excellent code quality, documentation, and ease of use
* Well-implemented approximation of data center cooling environment in Python
* Interesting heterogeneous multi-agent experiments
* Relevant to both data center management and RL research

Cons:
* Lack of validation against established modeling tools (e.g., EnergyPlus, Modelica)
* Possible overfitting in baseline training to the provided carbon intensity data
* Figure 4 (benchmarking algos) was somewhat confusing and/or less informative

**Strengths:**

* Operating data centers efficiently is becoming increasingly important.
* Excellent code quality and documentation. It was easy to install, train, evaluate, and visualize.
* The authors have carefully implemented a good-enough approximation of the data center cooling environment in Python, making this package easy to install and customize with Python-based RL libraries.
* The heterogeneous multi-agent baselines, Experiments 1 (single vs. multi-agent) and 2 (reward sharing) are particularly interesting for the RL community

**Additional Feedback:**

The manuscript has no line numbers, and the submission checklist is different from the current one.

**Documentation:**

Yes. Excellent code quality and documentation. It was easy to install, train, evaluate, and visualize.

**Limitations:**

Enabling control of the workload is great, but the current implementation might be too simple. The precise information might not be readily available, but it would be much more valuable if more information and/or realistic modeling about the workload were included, such as the proportion of flexible tasks and price/profit.

**Opportunities For Improvement:**

* It would be very convincing if the newly implemented data center cooling model were validated against modeling results from EnergyPlus (Sun et al., 2021) or Modelica (Zuo et al., 2021).
* A composite score like operating cost, which may account for the cost of electricity, water, carbon emission, and task penalty, would be helpful for quantifying and comparing the policies, and even optimizing them for different locations. Also, having baseline metrics from random and rule-based controllers would be helpful to set up a "lower bound".
* When training the baselines, was the carbon intensity data used as the observation for current and forecast grid CI data? If so, trained checkpoints might have overfitted to that particular dataset. Developing data-driven simulators for carbon intensity, workload, and weather, based on historical patterns, would be helpful to train more robust policies; this can also allow evaluation using the actual historical data.
* The authors claim that SustainDC is faster than the previous packages that rely on EnergyPlus or Modelica. It would be helpful if the simulation speed (i.e., steps per second) is provided (e.g., vs. Sinergym).

Questions:
* Experiments 3-6 used the same reward across different locations, varying only the algorithm. If the reward heavily favors CFP reduction, wouldn't all algorithms tend to decrease CFP? This raises questions about the value of comparing algorithms on these dimensions. Clarification on this experimental design would be helpful.
* Are there particular reasons to use only CPUs for training?

**Relation To Prior Work:**

* Yes.
* Zuo et al., 2021 should be cited.

Zuo, Wangda, Wetter, Michael, VanGilder, James, Han, Xu, Fu, Yangyang, Faulkner, Cary, Hu, Jianjun, Tian, Wei, and Condor, Michael. Improving Data Center Energy Efficiency Through End-to-End Cooling Modeling and Optimization. Final Report. United States: N. p., 2021. Web. doi:10.2172/1773506.

**Summary And Contributions:**

This work introduces SustainDC, an accessible Python-based platform for studying data center optimization using reinforcement learning. Key contributions include:
* Implemented multi-agent environments for data center operations fully in Python, encompassing workload scheduling, cooling optimization, and battery management. These environments are designed for easy use and customization
* Provided heterogeneous multi-agent baselines along with ablation and reward shaping experiments

---

> ### Author Rebuttal · Authors · 2024-08-16
>
> We sincerely thank reviewer XQ3a for the valuable and thoughtful feedback on different aspects of the work. Please see our responses below and let us know if you need any further clarification.
>
> "However, I think validation against EnergyPlus or Modelica is critical.:"
>
> We plan to add a benchmarking with respect to the EnergyPlus and Modelica. Currently, we compare the speed of simulation for SustainDC vs EnergyPlus for a data center with same number of server in the figure attached in the PDF. We will add further information on this topic to the open-source repository and documentation.
>
> "Also, additional features like random/rule-based baselines and carbon/weather data generators would complete the package.:"
>
> Our framework enables users to choose their own rule based baselines (e.g. we have used G36 [1] for cooling optimization and a real time version of analytical battery scheduling method [2]). We plan to augment our repository with additional baselines in the future. As a feature itself, SustainDC allows users to create random or any custom baseline they want from here. The user can also have carbon and weather data generators as outlined here for carbon intensity data and here for custom weather data. We plan to add this discussion to the paper itself.
>
> "Possible overfitting in baseline training to the provided carbon intensity data:"
>
> For some of our tests, the MARL agents were trained on one location and evaluated for different locations.
>
> "Figure 4 (benchmarking algos) was somewhat confusing and/or less informative:"
>
> We sincerely apologize for the potential confusion in figure 4. The purpose of these images were to motivate the use of RL agents in figure 4.a. Figure 4.b showed an ablation on the collaborative reward weight α. Figures 4.c to 4.d explains the difference in performance across different regions for independent MARL (IPPO), shared critic homogeneous MARL agent (MAPP) and a set of shared critic heterogenous MARL agents (HAPPO, HAA2C, HAD3QN, HASAC). We shall better explanation of this discussion in the results section of the paper.
>
> "Relation To Prior Work:"
>
> If accepted, we will cite the paper: Zuo, Wangda, Wetter, Michael, VanGilder, James, Han, Xu, Fu, Yangyang, Faulkner, Cary, Hu, Jianjun, Tian, Wei, and Condor, Michael. Improving Data Center Energy Efficiency Through End-to-End Cooling Modeling and Optimization. Final Report. United States: N. p., 2021. Web. doi:10.2172/1773506.
>
> [1]	K. Zhang, D. Blum, H. Cheng, G. Paliaga, M. Wetter, J. Granderson, Estimating ashrae guideline 36 energy savings for multi-zone variable air volume systems using spawn of energyplus, Journal of Building Performance Simulation 15 (2022) 215–236
>
> [2]	B. Acun, B. Lee, F. Kazhamiaka, K. Maeng, U. Gupta, M. Chakkaravarthy, D. Brooks, C.-J. Wu, Carbon explorer: A holistic framework for designing carbon aware datacenters, in: Proceedings of the 28th ACM International Conference on Architectural Support for Programming Languages and Operating Systems, Volume 2, ACM, 2023. URL: https://doi.org/10.1145/3575693.3575754. doi:10.1145/3575693.3575754

---

### Official Review · Reviewer_MtEx · 2024-07-20
**Sustain DC: A Pivotal RL Environment for Advancing Energy, Water, and Resource Efficiency in Data Centres**

**Rating:** 8
**Confidence:** 4

**Review:**

# Evaluation of SustainDC

## Quality

SustainDC demonstrates high-quality research in the field of sustainable data centre control using multi-agent reinforcement learning (MARL). The authors have developed a comprehensive and well-thought-out framework that addresses multiple aspects of data centre operations.

### Pros:
1. **Rigorous methodology**: The paper presents a detailed approach to modelling various components of data centre operations, including workload management, cooling systems, and battery storage.
2. **Extensive experimentation**: The authors conduct thorough benchmarking of various MARL algorithms across different data centre configurations and geographical locations.
3. **Reproducibility**: The provision of open-source code, documentation, and experimental settings enhances the reproducibility of the research.

### Cons:
1. **Limited real-world validation**: While the simulations are comprehensive, the paper lacks validation against real-world data centre operations.
2. **Absence of comparative baselines**: The paper could benefit from comparisons with traditional, non-ML control strategies to better quantify the improvements offered by MARL approaches.

## Clarity

The paper is generally well-written and structured logically, making it accessible to readers familiar with reinforcement learning and data centre operations.

### Pros:
1. **Clear structure**: The paper follows a logical flow, from introduction to methodology to results and conclusion.
2. **Detailed explanations**: The authors provide comprehensive descriptions of the SustainDC components and the benchmarking process.
3. **Effective use of figures**: The paper includes helpful visualisations, such as the overview of SustainDC (Figure 3) and the benchmark results (Figure 4).

### Cons:
1. **Technical complexity**: Some sections may be challenging for readers without a strong background in both reinforcement learning and data centre operations.
2. **Abbreviation overuse**: The frequent use of abbreviations (e.g., HVAC, CFP, CI) might hinder readability for some readers.

## Originality

SustainDC presents a novel contribution to the field of sustainable data centre control.

### Pros:
1. **Innovative integration**: The framework uniquely combines workload scheduling, cooling optimisation, and battery management in a single MARL environment.
2. **Customisable approach**: The high degree of customisation in SustainDC allows for testing various data centre configurations and control strategies, which is not common in existing frameworks.
3. **Multi-agent focus**: The emphasis on MARL, including support for heterogeneous agents, sets this work apart from previous single-agent approaches.

### Con:
1. **Building on existing concepts**: While the integration is novel, many of the individual components are based on existing concepts in data centre management and reinforcement learning.

## Significance

The significance of SustainDC in addressing the critical issue of data centre sustainability is substantial.

### Pros:
1. **Timely contribution**: With the increasing energy consumption of data centres, this work addresses a pressing need in the field of sustainable computing.
2. **Broad applicability**: The framework's flexibility allows it to be applied to a wide range of data centre designs and locations, enhancing its potential impact.
3. **Research catalyst**: By providing an open-source, customisable environment, SustainDC has the potential to accelerate research in sustainable data centre control.
4. **Practical implications**: The work has clear implications for improving energy efficiency and reducing carbon footprints in real-world data centres.

### Cons:
1. **Implementation challenges**: The paper does not thoroughly discuss the potential challenges in implementing these MARL strategies in real-world data centres.
2. **Long-term effects**: The long-term performance and stability of the proposed approaches over extended periods are not extensively explored.

## Mathematical Formulation

The authors provide a clear mathematical formulation of the multi-agent control problem. For example, the objective function for minimising carbon footprint (CFP) over a horizon $N$ is given as:

$$\left[\theta_{LS}, \theta_{DC}, \theta_{BAT}\right] = \argmin \left(\sum_{t=0}^N CFP_t\right)$$

where $\theta_{LS}$, $\theta_{DC}$, and $\theta_{BAT}$ are the parameters for the Load Shifting, Data Centre, and Battery agents respectively.

The carbon footprint at each time step is calculated as:

$$CFP_t = (E_{hvac} + E_{it} + E_{bat}) \times CI_t$$

where $E_{hvac}$, $E_{it}$, and $E_{bat}$ represent the energy consumption of HVAC, IT equipment, and battery respectively, and $CI_t$ is the grid carbon intensity at time $t$.

These formulations clearly define the optimization problem and the key components of the system, providing a solid foundation for the MARL approach.

## Conclusion

SustainDC represents a significant contribution to the field of sustainable data centre control. Its comprehensive approach, flexibility, and focus on MARL make it a valuable tool for researchers and practitioners. While there are areas for improvement, particularly in real-world validation and long-term performance analysis, the overall quality, originality, and potential impact of this work are high. The open-source nature of the project further enhances its value to the research community, potentially accelerating advancements in sustainable data centre operations.

**Strengths:**

## Significance of Contribution

SustainDC addresses a critical need in the field of sustainable computing by tackling the pressing issue of data centre sustainability. This is particularly significant given the exponential growth in computational demand and its environmental impact. The framework provides a much-needed platform for developing and evaluating advanced control strategies for data centres, offering a comprehensive approach that uniquely integrates multiple aspects of data centre operations.

By combining **workload scheduling**, **cooling optimisation**, and **battery management** into a single multi-agent reinforcement learning (MARL) environment, SustainDC enables a holistic approach to optimising data centre efficiency and sustainability. This integration allows researchers and practitioners to explore complex interactions between different operational components, potentially leading to more effective and efficient control strategies.

One of the key strengths of SustainDC is its *versatility and customisation capabilities*. The highly configurable environments allow for testing various data centre designs and operational scenarios, making it applicable to a wide range of real-world situations. Furthermore, its support for both homogeneous and heterogeneous MARL agents broadens its applicability and allows for exploration of diverse control strategies.

## Relevance to Broader Research Community

SustainDC's **interdisciplinary nature** is a significant strength, as it bridges the gap between machine learning, energy systems, and data centre operations. This makes it relevant to researchers across multiple fields, including AI, sustainability, and computer systems engineering. The potential for cross-pollination of ideas across these disciplines is substantial.

The *open-source nature* of SustainDC is another major strength. By providing code, documentation, and experimental settings, the authors promote reproducibility and collaboration within the research community. This openness serves as a valuable resource for both academic researchers and industry practitioners, potentially accelerating advancements in the field.

Furthermore, SustainDC offers a complex, real-world inspired environment for testing and developing MARL algorithms. This makes it an excellent **benchmark for the MARL community**, with potential applications extending beyond data centre control. The challenging nature of the environment could drive significant advancements in MARL algorithms and techniques.

## Quality of Research

The quality of research demonstrated in SustainDC is noteworthy. The authors have employed a **rigorous methodology**, presenting detailed modelling of data centre components based on established physical principles and industry standards. This attention to detail enhances the realism and reliability of the simulations.

The *comprehensive benchmarking* across various algorithms, configurations, and geographical locations further underscores the quality of the research. By incorporating real-world factors such as weather conditions, grid carbon intensity, and workload variations, SustainDC enables more accurate evaluation of control strategies compared to simplified models.

The **extensibility** of SustainDC is another strength in terms of research quality. Built on the Gymnasium Env class, it allows for easy implementation of new control strategies. The flexible reward formulation enables exploration of various optimisation objectives, making it a versatile tool for diverse research questions.

## Ethical and Social Implications

The ethical and social implications of SustainDC are significant and positive. By directly contributing to efforts in **reducing carbon emissions and energy consumption** of data centres, it has the potential to significantly decrease the environmental footprint of rapidly growing digital infrastructure. Moreover, it optimises not just energy use but also water consumption, addressing multiple aspects of sustainability.

The *open-source nature* of SustainDC promotes transparency in data centre operations and control strategies, encouraging collaborative efforts in developing sustainable computing solutions. This openness could lead to more rapid advancements and wider adoption of sustainable practices in the industry.

From an economic perspective, SustainDC has the potential to **reduce operational costs** of data centres, which could lead to more affordable digital services. This has broader societal implications in terms of access to digital resources and services.

Finally, SustainDC provides a sophisticated research platform that might otherwise be inaccessible due to the high costs and complexities of real data centre experimentation. This *democratisation of research capabilities* could lead to more diverse and innovative solutions in the field of sustainable computing.

SustainDC presents a significant contribution to the field of sustainable computing, offering a robust, versatile, and ethically aligned platform for advancing data centre control strategies. Its relevance spans multiple research communities and has the potential to drive meaningful progress in reducing the environmental impact of digital infrastructure.

**Additional Feedback:**

Technical Robustness and Clarity
The paper provides a comprehensive overview of the SustainDC framework, which is commendable. However, some sections could benefit from additional clarity and detail. For instance, the mathematical models and algorithms should be explained with more explicit notation and examples to enhance understanding. It would be helpful to include a flowchart or diagram that visualizes the overall process and interactions between the different components of the system. This could help readers grasp the operational flow and decision-making processes more intuitively.

Evaluation and Benchmarking
The evaluation section is thorough, yet it could be improved by including a more diverse set of benchmarks. Currently, the focus is on specific geographical locations and workload traces. Expanding the evaluation to include additional scenarios, such as different types of data centres (e.g., edge data centres) and varying scales of operations, could provide more comprehensive insights into the robustness and adaptability of the proposed solutions. Additionally, providing a more detailed analysis of the performance metrics and their implications would strengthen the overall evaluation.

Ethical and Social Implications
As previously mentioned, addressing ethical and social implications more thoroughly would enhance the paper's impact. Specifically, the potential for job displacement due to automation in data centre management is a significant concern. The authors should consider proposing strategies for mitigating this impact, such as upskilling programs for affected employees or creating new roles that leverage the enhanced capabilities of automated systems. Additionally, discussing the energy rebound effect and suggesting policies or measures to prevent increased demand from offsetting energy savings would be valuable.

**Clarity:**

The paper is well written and demonstrates good writing quality and clear organization. It follows a logical structure, starting with a clear introduction that establishes the context and motivation for the work. The authors effectively explain complex concepts related to data centre operations and multi-agent reinforcement learning, making the content accessible to readers from various backgrounds.
The methodology is presented in detail, with clear explanations of the different environments (Workload, Data Centre, and Battery) and their interactions.

The use of figures, particularly Figure 3, helps in visualizing the overall structure of SustainDC. The results section is comprehensive, presenting benchmark comparisons through well-designed radar charts that allow for easy comparison across multiple metrics. While some technical jargon is unavoidable given the subject matter, the authors generally provide sufficient context for understanding. Overall, the paper succeeds in conveying its main ideas and contributions clearly and effectively.

**Correctness:**

Based on the paper and supplementary materials provided, the claims made in the submission appear to be generally correct, and the benchmark (SustainDC) seems to be constructed in a sound manner. The evaluation methods and experimental design are appropriate for the task at hand. Here's a brief assessment:

1. Benchmark Construction: SustainDC is built on well-established principles of data centre operations, incorporating models for workload management, cooling systems, and battery storage. The use of real-world data for workload traces, weather patterns, and grid carbon intensity adds to its validity.

2. Evaluation Methods: The authors use appropriate metrics for evaluating the performance of different MARL algorithms, including carbon footprint, HVAC energy consumption, IT energy consumption, task queue length, and water usage. These metrics cover key aspects of data centre sustainability and efficiency.

3. Experiment Design: The experiments are conducted across various geographical locations and data centre configurations, which helps to test the generalizability of the proposed approaches. The comparison of different MARL algorithms (IPPO, MAPPO, HAPPO, HAA2C, HAD3QN, HASAC) is comprehensive and well-executed.

4. Results Reporting: The authors provide detailed results, including performance comparisons across different metrics and locations. They also include ablation studies on the effects of reward sharing, which adds depth to the analysis.

However, there are a few areas where additional information or clarification could strengthen the work:

1. The paper could benefit from more detailed comparisons with traditional, non-ML control strategies to better quantify the improvements offered by MARL approaches.

2. While the authors mention using reduced-order models for certain components (e.g., pumps and cooling towers), more information on the potential impact of these simplifications would be valuable.

3. A more in-depth discussion of the long-term stability and robustness of the proposed solutions would enhance the credibility of the results.

Overall, the benchmark appears to be constructed in a sound way, with appropriate evaluation methods and experimental design. The claims made are generally supported by the presented evidence, though some areas could benefit from additional validation or clarification.

**Documentation:**

Based on the paper and supplementary materials provided, SustainDC is primarily a benchmarking environment rather than a dataset submission. However, it does involve the use of various datasets for its simulations.

### Data Collection and Organization:

The authors mention using open-source workload traces from Alibaba and Google data centres.
Weather data in .epw format is sourced from energyplus.net for various locations.
Carbon intensity data is extracted from eia.gov.
The organization of these datasets within the SustainDC framework is explained.


### Availability and Maintenance:

The authors provide a GitHub repository link for the code and documentation.
They mention that all code, licenses, and instructions can be found on GitHub.
However, there's no explicit discussion of a long-term maintenance plan for the benchmark or associated datasets.


### Ethical and Responsible Use:

The paper doesn't explicitly discuss ethical considerations or guidelines for responsible use of the benchmark or underlying data.


### Reproducibility:

The supplementary material provides details on how to run the benchmark, including the required data format and file structure.
A default configuration dictionary is mentioned, which should aid in reproducibility.
The authors provide information on the computational resources used for their experiments.


### Documentation and Intended Uses:

The paper clearly outlines the intended use of SustainDC for benchmarking MARL algorithms in data center control.
A link to additional documentation is provided.


### Licensing:

- The SustainDC benchmark itself uses an MIT license, which is a permissive open-source license. This is a positive aspect for accessibility and reuse.
- The Google cluster workloads use a Creative Commons license, which is also favourable for research use.
- The Alibaba Cluster Trace Program lacking an explicit license is a potential concern.


### Areas for improvement:

- Provide more detailed information on data pre-processing steps, especially for the workload traces and carbon intensity data.
- Include explicit guidelines for ethical and responsible use of the benchmark and associated data.
- Outline a long-term maintenance plan for the benchmark and datasets.
- Clearly state the licensing terms for both the benchmark code and any included datasets.
- Consider providing a specific URL for reviewer access to the datasets used in the benchmark, if possible.

While the authors provide substantial information to support reproducibility of their benchmark, there are areas where more detail could be provided, particularly regarding ethical considerations, and long-term maintenance plans.

**Ethics:**

The submission does not involve human subjects directly, thus eliminating concerns related to human research ethics. The primary datasets from Alibaba and Google are publicly available, but the paper should verify compliance with data privacy regulations and proper attribution. Potential biases and limitations of these datasets must be addressed to ensure the validity and generalizability of the research findings. The paper should discuss any safety and security risks posed by the reinforcement learning algorithms and ensure they do not introduce vulnerabilities in data centre operations.

While the paper aims to optimize energy consumption and reduce the carbon footprint of data centres, it does not consider the potential for an energy rebound effect, where increased efficiency might lead to higher demand for data services. The complexity of the proposed system might also create barriers to entry for researchers or organizations with limited resources, despite its open-source nature. Also there is potential for job displacement due to automation in data centre management is a significant social implication that is not thoroughly addressed.

Privacy concerns related to using detailed operational data for training RL agents are not extensively discussed. Additionally, highly optimized data centres could be misused for activities with negative societal impacts, such as surveillance or cryptocurrency mining. The paper should explore these potential misuses and suggest safeguards to ensure responsible use.

Overall, while there are minor ethical concerns, addressing these issues will strengthen the submission and I do not see it conflicting with any NeurIPS ethical guidelines.

**Limitations:**

The authors have partially addressed limitations and potential negative societal impacts of their work, but there is room for improvement. They mention some limitations, such as the absence of an oracle for optimal results and the use of reduced-order models for certain components. However, a more comprehensive discussion of limitations would strengthen the paper.

To improve, the authors could address the simulation-reality gap, discuss potential privacy concerns related to data centre operational data, and explore the long-term stability of their proposed solutions. They should also consider broader societal impacts, such as potential job displacement due to automation and the energy rebound effect. Including a dedicated "Limitations and Future Work" section would demonstrate a more thorough understanding of their work's scope and potential drawbacks. Additionally, discussing how SustainDC might be validated in real-world settings would enhance the paper's practical relevance. By being more upfront about these aspects, the authors would provide a more balanced view of their contribution and offer valuable insights for future research directions.

**Opportunities For Improvement:**

## Significance of Contribution

While SustainDC offers a comprehensive approach to data centre sustainability, there are some limitations to its contribution:

1. **Simulation vs. Reality Gap**: The framework, while detailed, is still a simulation. There may be discrepancies between the simulated environment and real-world data centres that could affect the direct applicability of the developed strategies.

2. **Scope of Sustainability**: Although SustainDC covers energy, cooling, and workload management, it may not address all aspects of data centre sustainability. For instance, it doesn't appear to consider the lifecycle of hardware or the environmental impact of data centre construction.

3. **Generalizability**: The effectiveness of strategies developed using SustainDC may vary significantly when applied to data centres with very different architectures or in extreme geographical locations not covered in the current version.

## Relevance to Broader Research Community

Despite its interdisciplinary nature, there are some limitations in terms of broader relevance:

1. **Learning Curve**: The complexity of the environment might present a steep learning curve for researchers not familiar with both reinforcement learning and data centre operations, potentially limiting its adoption.

2. **Computational Requirements**: The paper doesn't thoroughly discuss the computational resources required to run simulations, which could be a barrier for researchers with limited access to high-performance computing resources.

3. **Validation in Industry**: While relevant to both academia and industry, the paper doesn't present evidence of industry validation or adoption, which could limit its immediate impact on real-world data centre operations.

## Quality of Research

Although the research quality is high, there are some limitations:

1. **Lack of Comparison to Traditional Methods**: The paper focuses on comparing different RL algorithms but doesn't extensively compare the MARL approach to traditional, non-ML control strategies used in data centres.

2. **Limited Long-term Performance Analysis**: The paper doesn't thoroughly explore the long-term stability and performance of the proposed approaches, which is crucial for real-world application.

3. **Sensitivity Analysis**: There's limited discussion on how sensitive the results are to changes in various parameters, which could affect the robustness of the developed strategies.

## Ethical and Social Implications

While the ethical and social implications are largely positive, there are some potential limitations:

1. **Privacy Concerns**: The paper doesn't extensively discuss potential privacy implications of using detailed operational data for training RL agents.

2. **Job Displacement**: The potential for automation in data centre management could lead to job displacement, a social implication not thoroughly addressed.

3. **Energy Rebound Effect**: While aiming to reduce energy consumption, more efficient data centres could lead to increased demand for data services, potentially offsetting some of the energy savings (known as the rebound effect).

4. **Equity in Access**: While the open-source nature is commendable, the potential complexity of the system might still create barriers to entry for researchers or organizations with limited resources.

5. **Potential for Misuse**: Highly optimized data centres could be used for purposes that have negative societal impacts (e.g., surveillance, cryptocurrency mining), a consideration not deeply explored in the paper.

While SustainDC presents a significant advancement in the field of sustainable data centre management, addressing these limitations could further enhance its impact and applicability in real-world scenarios. Future work could focus on bridging the simulation-reality gap, exploring longer-term performance, and more deeply considering the broader ethical and social implications of highly optimized data centres.

**Relation To Prior Work:**

The paper does discuss how SustainDC differs from previous contributions, but the comparison could be more comprehensive and explicit.

The authors address the relation to prior work primarily in Section 2 (Related Work). They highlight some key differences:

1. Implementation: Unlike previous environments that rely on EnergyPlus or Modelica, SustainDC is implemented entirely in Python, allowing for easier customization and faster execution.

2. Customizability: The authors emphasize SustainDC's high degree of customization, allowing users to test fine-grained design choices across various DC components.

3. Focus on MARL: SustainDC is specifically designed for benchmarking multi-agent reinforcement learning algorithms in the context of data centres, which is not a primary focus of previous works.

4. Comprehensive modelling: The framework integrates workload scheduling, cooling optimization, and battery management, providing a more holistic approach than many existing solutions.

However, there are areas where the discussion of differences could be improved:

1. Direct comparisons: The paper would benefit from more explicit, point-by-point comparisons with specific previous works, highlighting exactly what SustainDC can do that others cannot.

2. Limitations of previous approaches: While the authors mention some limitations of existing tools (e.g., cross-platform latency issues), a more detailed discussion of how SustainDC overcomes these specific challenges would strengthen the paper.

3. Quantitative comparisons: If possible, including quantitative comparisons (e.g., simulation speed, level of detail in modelling) with existing frameworks would more clearly illustrate SustainDC's advantages.

4. Novel aspects: While the integration of various components is highlighted as novel, the authors could more clearly articulate which specific aspects of their modelling or approach are entirely new contributions to the field.

While the paper does discuss how SustainDC differs from previous contributions, there is room for a more comprehensive and explicit comparison. Enhancing this aspect would more effectively highlight the novelty and significance of SustainDC in the field of sustainable data centre control.

**Summary And Contributions:**

This submission introduces SustainDC, a comprehensive set of Python environments for benchmarking multi-agent reinforcement learning (MARL) algorithms in sustainable data centre control. The main contributions are:

1. **Customisable environments**: SustainDC provides highly configurable simulations of data centre operations, allowing researchers to test various designs and configurations.

2. **Multi-agent RL focus**: The framework supports MARL controllers with both homogeneous and heterogeneous agents, as well as non-ML controllers, facilitating research into complex, coordinated control strategies.

3. **Holistic approach**: SustainDC integrates multiple aspects of data centre management, including workload scheduling, cooling optimisation, and battery management, enabling a system-wide approach to sustainability.

4. **Realistic modelling**: The environments incorporate real-world factors such as weather conditions, grid carbon intensity, and workload variations across different geographical locations.

5. **Extensible framework**: Built on the Gymnasium Env class, SustainDC allows for easy implementation and comparison of various control strategies.

6. **Comprehensive benchmarking**: The authors provide extensive evaluations of different MARL algorithms across diverse data centre designs and locations.

7. **Open-source contribution**: The codebase, documentation, and experimental settings are made publicly available, promoting reproducibility and further research in the field.

SustainDC addresses the critical need for sustainable computing in the face of increasing data centre energy consumption. By providing a sophisticated RL environment that considers energy use, water consumption, and overall resource efficiency, it offers a valuable tool for researchers and practitioners working towards more environmentally responsible data centre operations.

---

> ### Author Rebuttal · Authors · 2024-08-16
>
> We thank reviewer MtEx for the thorough, deeply insightful and helpful feedback on the paper and address some of your comments.
>
> "Limited real-world validation: While the simulations are comprehensive, the paper lacks validation against real-world data centre operations":
>
> We are planning to deploy the trained agents to real data centers. We are working towards domain adaptation for deployment with safeguards. We will augment the codebase with these updates.
>
> "Absence of comparative baselines: The paper could benefit from comparisons with traditional, non-ML control strategies to better quantify the improvements offered by MARL approaches.":
>
> Our baselines for the cooling setpoint optimization and the battery storage optimization comparisons involve an implementation of the ASHRAE G36 standard [1] reset request based set point changes, and we adopted the analytical battery energy storage scheduler from the paper Carbon Explorer [2] and modified it for real time decision-making (i.e.: we reduced the optimization horizon in the original paper from 24 hours to 3 hours). In future work, we shall include further basline comparisons using MPCs and other non-ML control algorithms.
>
> [1]	K. Zhang, D. Blum, H. Cheng, G. Paliaga, M. Wetter, J. Granderson, Estimating ashrae guideline 36 energy savings for multi-zone variable air volume systems using spawn of energyplus, Journal of Building Performance Simulation 15 (2022) 215–236
> [2]	B. Acun, B. Lee, F. Kazhamiaka, K. Maeng, U. Gupta, M. Chakkaravarthy, D. Brooks, C.-J. Wu, Carbon explorer: A holistic framework for designing carbon aware datacenters, in: Proceedings of the 28th ACM International Conference on Architectural Support for Programming Languages and Operating Systems, Volume 2, ACM, 2023. URL: https://doi.org/10.1145/3575693.3575754. doi:10.1145/3575693.3575754
>
> "Implementation challenges: The paper does not thoroughly discuss the potential challenges in implementing these MARL strategies in real-world data centres.:"
>
> Thank you for pointing out this important aspect of the work. We will add this information to the open source repository associated with the paper.
>
> Implementation challenges for real data centers: In order to have a smooth integration with current systems where HVAC runs in isolation, we plan a phased deployment with recommendation to the data center operative followed by direct integration of the control agents with the HVAC system with safeguards. For real-world deployment, a trained model should be run on a production server using appropriate checkpoints within a containerized platform with necessary dependencies. Security measures must restrict the software to only read essential data, generate decision variables, and write them with limited access to secure memory for periodic reading by the data center’s HVAC management system. To ensure robustness against communication loss, a backup mechanism for generating decision variables is essential.
>
> "Long-term effects: The long-term performance and stability of the proposed approaches over extended periods are not extensively explored.:"
>
> To plan for equipment changes and data center accessories, we will implement continual reinforcement learning to avoid out-of-distribution errors.
>
> "Significance of Contribution:"
>
> To bridge the reality vs simulation gap, we are working on a Digital Twin for a data center, which is our ultimate goal. We are working on expanding the scope of SustainDC to include other aspects of data center sustainability, such as hardware lifecycle and construction impact (sustainable manufacturing). To address generalizability, we have currently considered different locations with wide variation of temperature and grid carbon intensity.
> Relevance to Broader Research Community: Validation in Industry We are working on realizing some of the proposed methods with consortiums like ExaDigiT which focuses on HPC and Supercomputing along with other customers and industry collaborators.

---

> > ### Comment · Reviewer_MtEx · 2024-08-22
> >
> > Based on the feedback, it demonstrated the authors have plans for addressing potential limitations but also collecting real world evidence to enhance the simulation. Continuous RL may be potential option for the improvement of the benchmark, but I am not sure if it will solve the underlying challenges related to equipment in the long term. This however is open to debate, and I find the responses satisfactory.

---

### Decision · Program_Chairs · 2024-09-26

**Decision:**

Accept (Poster)

**Comment:**

This work presents a novel framework designed to benchmark multi-agent reinforcement learning (MARL) algorithms in sustainable data center control. The primary contributions of SustainDC include its customizable and realistic data center simulations, support for both MARL and non-ML controllers, and a holistic integration of key operational aspects like workload scheduling and cooling optimization. Its extensible framework, built on the Gymnasium Env class, facilitates comparative research and extends the potential for innovation in sustainable data center management. Furthermore, the open-source nature of SustainDC ensures its broad applicability and promotes reproducibility, making it a valuable asset for ongoing research in sustainable computing.